Description and phylogenetic relationships of a new species of Torvoneustes (Crocodylomorpha, Thalattosuchia) from the Kimmeridgian of Switzerland

Girard Léa C. 1 2 lea.girard@unifr.ch
De Sousa Oliveira Sophie 2
Raselli Irena 1 3
Martin Jeremy E. 4
http://orcid.org/0000-0001-5539-8691 Anquetin Jérémy 1 3
1 Department of Geosciences, University of Fribourg , Fribourg , Switzerland
2 Géosciences Rennes, Université Rennes I , Rennes , France
3 Jurassica Museum , Porrentruy , Switzerland
4 Laboratoire de Géologie de Lyon, Terre, Planète, Environnement, UMR CNRS 5276 (CNRS, ENS, université Lyon 1), Ecole Normale Supérieure de Lyon , Lyon , France
De Baets Kenneth
Electronic publication date: 2023 Jul 19
Publication date: 2023
Volume: 11
Electronic Location ID: e15512
Received 2022 Dec 23; Accepted 2023 May 15
Copyright: © 2023 Girard et al.
Copyright year: 2023
Copyright holder: Girard et al.
License: This is an open access article distributed under the terms of the Creative Commons Attribution License, which permits unrestricted use, distribution, reproduction and adaptation in any medium and for any purpose provided that it is properly attributed. For attribution, the original author(s), title, publication source (PeerJ) and either DOI or URL of the article must be cited.
License URL: https://creativecommons.org/licenses/by/4.0/

Keywords: Crocodylomorpha, Thalattosuchia, Metriorhynchidae, Vertebrate palaeontology, Jurassic, Switzerland, Kimmeridgian

Funding: Swiss-European Mobility Programme (University of Fribourg) Swiss National Science Foundation SNF 205321_175978 Federal Roads Office Republic and Canton of Jura Sophie De Sousa Oliveira and Léa C Girard were supported by a grant from the Swiss-European Mobility Programme (University of Fribourg). This work was funded by a grant from the Swiss National Science Foundation (SNF 205321_175978) to Jérémy Anquetin and Irena Raselli. The Paleontology A16 project was funded by the Federal Roads Office (FEDRO, 95%) and the Republic and Canton of Jura (RCJU, 5%). The funders had no role in study design, data collection and analysis, decision to publish, or preparation of the manuscript.

==============================
Metriorhynchids are marine crocodylomorphs found across Jurassic and Lower Cretaceous deposits of Europe and Central and South America. Despite being one of the oldest fossil families named in paleontology, the phylogenetic relationships within Metriorhynchidae have been subject to many revisions over the past 15 years. Herein, we describe a new metriorhynchid from the Kimmeridgian of Porrentruy, Switzerland. The material consists of a relatively complete, disarticulated skeleton preserving pieces of the skull, including the frontal, prefrontals, right postorbital, nasals, maxillae, right premaxillae and nearly the entire mandible, and many remains of the axial and appendicular skeleton such as cervical, dorsal, and caudal vertebrae, ribs, the left ischium, the right femur, and the right fibula. This new specimen is referred to the new species Torvoneustes jurensis sp. nov. as part of the large-bodied macrophagous tribe Geosaurini. Torvoneustes jurensis presents a unique combination of cranial and dental characters including a smooth cranium, a unique frontal shape, acute ziphodont teeth, an enamel ornamentation made of numerous apicobasal ridges shifting to small ridges forming an anastomosed pattern toward the apex of the crown and an enamel ornamentation touching the carina. The description of this new species allows to take a new look at the currently proposed evolutionary trends within the genus Torvoneustes and provides new information on the evolution of this clade.

Introduction

Thalattosuchia Fraas, 1901 is a clade of mostly marine crocodylomorphs that lived from the Early Jurassic to the Early Cretaceous and had a near global distribution from the eastern margins of the Tethys, the opening Atlantic Ocean, down to the coasts of South America, China and north of Africa (Fraas, 1901, 1902; Andrews, 1913; Gasparini & Iturralde-Vinent, 2001; Fara et al., 2002; Frey et al., 2002; Buchy et al., 2006b; Chiarenza et al., 2015; Herrera et al., 2015; Johnson, Young & Brusatte, 2020; Young et al., 2020a; Wilberg et al., 2022). Thalattosuchians are subdivided into the more coastal Teleosauroidea Geoffroy Saint-Hilaire, 1831, and the fully aquatic Metriorhynchoidea Fitzinger, 1843. Among archosaurs, Metriorhynchidae show many skeletal, soft tissue and neuroanatomical features that suggest they were suited for a pelagic lifestyle, including: limbs transformed into flippers with a great reduction of the forelimbs and a simplification of the pelvic girdle; lengthening of the body; loss of osteoderms; a smooth skin; a hypocercal tail; a slender and lighter skull; hypertrophied salt glands; orbits placed laterally and overhung by the prefrontals; loss of the mandibular fenestra (Fraas, 1902; Andrews, 1913; Vignaud, 1995; Frey et al., 2002; Gandola et al., 2006; Pierce, Angielczyk & Rayfield, 2009; Young & Andrade, 2009; Young et al., 2010, 2020a, 2020b; Herrera, Fernández & Gasparini, 2013; Herrera et al., 2015; Foffa et al., 2018; Sachs et al., 2021; Spindler et al., 2021; Le Mort et al., 2022; Schwab et al., 2022).

Metriorhynchids include two subclades. Metriorhynchinae Lydekker, 1889 are usually characterized by a slender body, an elongated snout, and a higher count of poorly ornamented teeth (Parrilla-Bel et al., 2013; Sachs et al., 2021). In contrast, Geosaurinae Lydekker, 1889 are more robust macrophagous predators with shorter snouts and a reduced number of large, often ornamented, ziphodont teeth (Young et al., 2012b, 2015). During the Middle Jurassic, each of these groups saw the emergence of the more derived tribes Rhacheosaurini and Geosaurini, respectively (Young et al., 2013a, 2013b, Foffa & Young, 2014, Foffa et al., 2017). The diversity of metriorhynchids has long been underestimated, but intensives revisions in the last two decades and the description new material significantly improved the knowledge of the group and its inner phylogenetic relationship (Frey et al., 2002; Wilkinson, Young & Benton, 2008; Pierce, Angielczyk & Rayfield, 2009; Young & Andrade, 2009; Andrade et al., 2010; Wilberg, 2012; Young et al., 2010, 2020a, 2020b; Cau & Fanti, 2011).

Compared to their counterparts in England or Germany, the Late Jurassic metriorhynchid fossil record of Switzerland is relatively poor. Most of the fossils of Swiss thalattosuchians belongs to teleosauroids (Krebs, 1967; Guignard & Weidmann, 1977; Rieppel, 1981) with rare occurrence of metriorhynchoids (Schaefer et al., 2018; Abel, Sachs & Young, 2020; Young et al., 2020b). In addition to the specimen described herein, several specimens of Thalattosuchia have been excavated in the canton of Jura in the past 20 years. They are mainly represented by isolated teeth, but a few well preserved teleosaurid skulls and skeletons are known (Schaefer, 2012; Schaefer et al., 2018). Unidentified remains of metriorhynchids have been found in the canton of Jura in the form of a nasal, frontal, femur and vertebrae, as well as a single tooth of Dakosaurus Quenstedt, 1856 (Schaefer et al., 2018). A metriorhynchid anterior rostrum is also known from the lower Tithonian of Bern representing an indeterminate Rhacheosaurini (Rieppel, 1979; Young et al., 2020b). Here we describe a new specimen from the upper Kimmeridgian of the canton of Jura, Switzerland (Fig. 1). It consists of a relatively complete, associated skeleton with many cranial and postcranial bones preserved. This specimen is identified herein as a new species of the genus Torvoneustes Andrade et al., 2010 and located in the phylogenetic context of Metriorhynchidae. This material also allows a reassessment of the evolutionary trends previously proposed for the genus.

Figure 1 Geographical map of the Ajoie region, Canton of Jura, Switzerland.

The excavation site of MJSN BSY008-465 (BSY, Courtedoux–Bois de Sylleux) is indicated along the A16 Transjurane highway (in grey).

Materials and Methods

Material

MJSN BSY008-465 is a disarticulated metriorhynchid skeleton (Fig. 2). The specimen was initially collected by the Pal A16 on a large block of limestone. Bones were prepared directly on the surface and kept on pedestals of rock. This initial phase of preparation, especially the acid preparation, was poorly controlled and resulted in damages of the more fragile bony elements such as some cranial and mandibular elements. More recently, all bones were completely removed from the block to facilitate scientific study. The preserved remains of MJSN BS008-465 include cranial and mandibular elements as well as material from the axial and appendicular skeleton. Many elements are fragmented and show evidence of deformation.

Figure 2 Taphonomical arrangement of the metriorhynchid skeleton MJSN BSY008-465.

(A) Photograph of the skeleton still embedded in the limestone block (see text); (B) drawing of the bones in their taphonomical position.

The preserved elements of the cranium include the frontal, prefrontals, right postorbital, nasals, maxillae, right premaxillae. The mandible is almost complete and preserve both angulars, surangulars, articulars, splenials, and dentaries. Many isolated teeth are also preserved. The postcranial elements include cervical, dorsal, and caudal vertebrae, ribs, the left ischium, the right femur, and the right fibula. Numerous bone fragments are not identifiable.

Geological setting

Between 2000 and 2011, controlled paleontological excavations were conducted by the Pal A16 before the construction of the A16 Transjurane highway in the canton of Jura (Fig. 1). They revealed the presence of several rich fossiliferous horizons of marls and limestones, as well as many dinosaurs tracksites in the Ajoie region around the city of Porrentruy (Marty et al., 2003). During the Late Jurassic, this region was part of a carbonate platform with diversified shallow depositional environments such as lagoons, reefs, channels and littoral zones forming layers rich in marine fossils (Colombié & Strasser, 2005; Comment et al., 2015).

MJSN BSY008-465 was found in 2008 on the hardground level 4,000 of the Lower Virgula Marls (Fig. 3) in the locality Courtedoux-Bois de Sylleux, Switzerland (Fig. 1). The Lower Virgula Marls belong to the Chevenez Member of the Reuchenette Formation. They date to the late Kimmeridgian and correspond to the end of the Mutabilis ammonite zone and beginning of the Eudoxus ammonite zone (Comment et al., 2015). These marls are notably characterized by the abundance of the small oyster Nanogyra virgula Koppka, 2015, which gives them their name. The hardground level 4,000 is rich in invertebrates, notably encrusted and benthic bivalves and brachiopods. Vertebrates are mostly represented by isolated material, with the exception of the metriorhynchid MJSN BSY008-465 described herein, a partial teleosauroid skeleton provisionally referred to Steneosaurus cf. bouchardi Sauvage, 1872, now combined as Proexochokefalos cf. bouchardi (Schaefer et al., 2018; Johnson, Young & Brusatte, 2020), and a relatively complete shell of the thalassochelydian turtle Thalassemys bruntrutana Puntener, Anquetin, and Billon-Bruyat, 2015 (Püntener, Anquetin & Billon-Bruyat, 2015).

Figure 3 Stratigraphical section of the Reuchenette Formation in Ajoie, Canton of Jura, Switzerland, with a close-up on the Lower Virgula Marls.

MJSN BSY008-465 was found on the hardground level 4,000 in the Lower Virgula Marls. The stratigraphical chart is derived from Comment et al. (2015) and Püntener, Anquetin & Billon-Bruyat (2020).

Phylogenetic analyses

The phylogenetic analyses were conducted using the data matrix and procedure of Young et al. (2020a), which were derived from Young et al. (2020b). The phylogenetic analysis has been performed before the publication of Sachs et al. (2021) which present an updated matrix from Young et al. (2020a). While we acknowledge this more recent matrix, the changes compared with the matrix of Young et al. (2020a) mainly focus on pneumaticity of cranial bones, acquired on CT data or broken bones. These characters are not applicable to the described specimen. Therefore using the revised matrix would not significantly improve the results within the frame of this study. The original matrix includes 179 taxa coded for 574 characters. The outgroup is Batrachotomus kupferzellensis Gower, 1999. The matrix was modified with Mesquite 3.61 (Maddison & Maddison, 2019) to include MJSN BSY008-465 as a new operational taxonomic unit (see Material S1). The latter was scored for 204 characters. The analytical procedure strictly follows the one described by Young et al. (2020a). The parsimony analyses were conducted using TNT 1.5 (No taxon limit) (Goloboff, Farris & Nixon, 2008; Goloboff & Catalano, 2016) with the RAM increased to 900 Mb. The analysis used the scripts provided by Young et al. (2020a) and consisted of an unweighted analysis followed by seven different extended implied weighting analyses (k = 1, 3, 7, 10, 15, 20 and 50). The main script (EIW.run) runs an initial “new technology” search (xmult:hits 10 replications 100 rss css xss fuse 5 gfuse 10 ratchet 20 drift 20; sec:drift 10 rounds 10 fuse 3; ratchet:numsubs 40 nogiveup; drift:numsubs 40 nogiveup) holding 20,000 trees per analysis, then runs a “traditional methods” search (bbreak:TBR) on the saved trees. The script then computes the descriptive statistics, the strict consensus tree, the majority rule consensus tree, and the maximum agreement subtree (for more details, see Young et al., 2020a).

The Bayesian analysis was conducted using MrBayes3.2.7 (Ronquist et al., 2012), again following the procedure described by Young et al. (2020a). The sampling model is a Markov Chain Monte Carlo with only variable characters scored and a Gama distribution (Mkv+G model). Five independent analyses are run, each with 10 chains for 10 million generations with a sampling every 5,000 generations and a burn-in of 40% (for more details, see Young et al., 2020a). At the end of the analysis, the consensus tree is saved.

Imaging

Each element of the cranium of MJSN BSY008-465 was individually scanned with a portable surface scanner (Artec Space Spider). The scans were treated with the software Artec Studio 13 to produce textured 3D models. The elements were assembled in Blender 2.8 in order to reconstruct the skull of MJSN-BSY008-465 in 3D (De Sousa Oliveira et al., 2023). This technique helped the description of the specimen and highlighted lost contacts between the bones. The 3D models of each individual element, as well as the reconstructed skull and mandible are made openly available in De Sousa Oliveira et al. (2023) and in Material S2.

The microscopic observation and detailed photographs of the teeth of MJSN BSY008-465 were realized with a digital microscope (Keyence VHX-970F).

The electronic version of this article in Portable Document Format (PDF) will represent a published work according to the International Commission on Zoological Nomenclature (ICZN), and hence the new names contained in the electronic version are effectively published under that Code from the electronic edition alone. This published work and the nomenclatural acts it contains have been registered in ZooBank, the online registration system for the ICZN. The ZooBank LSIDs (Life Science Identifiers) can be resolved and the associated information viewed through any standard web browser by appending the LSID to the prefix http://zoobank.org/. The LSID for this publication is: act:5DEFCF6F-D7EF-4711-9CB6-A7219F612ECB. The online version of this work is archived and available from the following digital repositories: PeerJ, PubMed Central SCIE and CLOCKSS.

Results

Systematic paleontology

CROCODYLOMORPHA Hay, 1930

THALATTOSUCHIA Fraas, 1901

METRIORHYNCHIDAE Fitzinger, 1843

GEOSAURINAE Lydekker, 1889

GEOSAURINI Lydekker, 1889

Torvoneustes Andrade et al., 2010

Diagnosis. see in Young et al. (2013b)

Type species. Dakosaurus carpenteri Wilkinson, Young & Benton, 2008

Included valid species. Torvoneustes coryphaeus Young et al., 2013b; Torvoneustes mexicanus Wieland, 1910; Torvoneustes jurensis sp. nov.

Occurrence. Kimmeridgian of Mexico (Barrientos-Lara et al., 2016); Kimmeridgian of Dorset, UK (Grange & Benton, 1996; Wilkinson, Young & Benton, 2008; Young et al., 2013b); middle Oxfordian to Tithonian of Oxfordshire, UK (Young, 2014); middle Oxfordian of Yorkshire (Foffa, Young & Brusatte, 2018); late Kimmeridgian of canton of Jura, Switzerland (Schaefer et al., 2018); upper Valanginian of Moravian-Silesian Region, Czech Republic (Madzia et al., 2021).

Torvoneustes jurensis sp. nov.

urn:lsid:zoobank.org:act:5DEFCF6F-D7EF-4711-9CB6-A7219F612ECB

Diagnosis. Torvoneustes jurensis sp. nov. is identified as a member of Torvoneustes by the following combination of characters: robust teeth lingually curved, conical crown, bicarinate with a prominent keel; enamel ornamentation made of conspicuous subparallel apico-basal ridges on the basal two thirds of the crown shifting to short, low-relief ridges forming an anastomosed pattern on the apex; carinae formed by a keel and true microscopic denticles forming a continuous row on the distal and mesial carinae; enamel ornamentation extending up to the carinae near the crown apex; inflexion point on the lateral margin of the prefrontals directed posterolaterally in dorsal view, at an angle of approximately 70° from the anteroposterior axis of the skull; acute angle (close to 60°) between the posteromedial and the lateral processes of the frontal. Torvoneustes jurensis differs from the other species of Torvoneustes by the following combination of characters: presence of a distinct angle between the anteromedial and the lateral processes of the frontal; lack of cranium ornamentation aside from faint ornamentation on the anterior part of the maxilla. Torvoneustes jurensis also differs from To. carpenteri in: lacking “finger-like” projections on the posterior margin of the prefrontal; having an angle of 142° in average between the anterior and lateral processes of the frontal; and having a long anteromedial process of the frontal reaching the same level as the anterior margin of the prefrontals (shorter process in To. carpenteri). Torvoneustes jurensis is also clearly distinct from To. coryphaeus in having: a tooth enamel ornamentation extending up to the carinae; prefrontals with a rounded distal margin (forming an acute angle in To. coryphaeus); and a supraorbital notch of 90° (45° in To. coryphaeus). The slender morphology, acuteness, and great number of teeth of Torvoneustes jurensis nov. sp. differs from both To. coryphaeus and To. carpenteri. Torvoneustes jurensis differs from To. mexicanus in having irregular denticle basal length (120–200 µm, mean: 160 µm) and distribution (density up to 40 denticles/5 mm in the upper middle part of the crown and down to 30 denticles/5 mm in the base and apex) on the carinae (regular basal length of 142 µm and density of 30 denticles/5 mm in To. mexicanus).

Etymology. The species is named after the Jura, which corresponds to both the mountain range and the Swiss canton where the holotype was found. The complete name could therefore be translated from Latin as “savage swimmer from Jura”.

Holotype. MJSN BSY008-465, a relatively complete disarticulated skeleton including the frontal, prefrontals, right postorbital, nasals, maxillae, right premaxillae, angulars, surangulars, articulars, splenials, dentaries, several isolated teeth, cervical, dorsal, and caudal vertebrae, ribs, the left ischium, the right femur, and the right fibula.

Type horizon and locality. Courtedoux-Bois de Sylleux, Ajoie, canton of Jura, Switzerland (Fig. 1). Lower Virgula Marls, Chevenez Member, Reuchenette Formation, late Kimmeridgian (Eudoxus ammonite zone), Late Jurassic (Fig. 3; Comment et al., 2015).

Description

Cranium

Cranial elements are disarticulated except for the frontal preserving its contact with the left prefrontal. The preserved elements of the cranium are exclusively part of the skull roof and snout. Many pieces are broken, incomplete, and/or deformed. As a result, the general shape and size of most of the skull fenestrae and apertures, such as the preorbital fossae, orbits, and supratemporal fenestrae cannot be clearly determined. Based on the skull reconstruction (Fig. 4), the skull total length is estimated at 88 cm, and the rostrum length (nasals and maxillae) is about 49 cm (55.7% of the total skull length), corresponding to the mesorostrine condition as defined by Young et al. (2010). The skull width-to-length ratio is 0.28. The supratemporal fenestrae are greatly enlarged and the orbits are facing laterally (Fig. 4).

Figure 4 MJSN BSY008-465, holotype of Torvoneustes jurensis (Kimmeridgian, Porrentruy, Switzerland).

Scientific drawings of the reconstructed skull (A) and mandible (B) in dorsal and (C) lateral views. the remaining matrix is represented in yellow. Anterior is to the right. Abbreviations: al, alveolus; an, angular; art, articular; cp, coronoid process; den, dentary; fr, frontal; mx, maxilla; na, nasal; pa, parietal; pmx, premaxilla; po, postorbital; prf, prefrontal; san, surangular; sdg, surangulodentary groove; sq, squamosal; spl, splenial; tc, tooth crown.

Premaxillae

An incomplete right premaxilla is preserved. Its anterior part is severely damaged and the posterior part, including the contact with the maxilla, is missing. The premaxilla seemingly suffered deformation as the location of the alveoli relative to the narial opening seems unnatural, looking more laterally placed than it should. The external nares aperture is partly conserved, oblong, moderate in width and formed entirely by the premaxilla (Figs. 5D, 5E). A complete alveolus, likely P2, is visible in ventral view. In dorsal view, the external surface of the bone is smooth except for a few superficial pits. The bone thickens on the external edge of the narial opening and close to the alveolus, but it thins posteriorly. The alveolar orientation suggests a tooth implantation directed anteroventrally (Fig. 4C).

Figure 5 MJSN BSY008-465, holotype of Torvoneustes jurensis (Kimmeridgian, Porrentruy, Switzerland).

(A) Right premaxilla, (B) right prefrontal, (C) left nasal, and (D) right nasal in dorsal views. Anterior is to the right.

Maxillae

Both maxillae are preserved. The right maxilla is almost complete, only missing its anteriormost part and the posterior end of the tooth row (Fig. 6). This element is heavily fractured and deformed, especially in its anterior part where the bone was broken in several parts and glued back together. This anterior part is deformed as the alveoli show extreme anteroposterior orientation that does not seems natural compared to the alveoli posterior to this major break. The remaining matrix prevents the observation of the medial side of the right maxilla. The left maxilla is incomplete and broken in four parts. Some of the fragments are severely damaged. Both maxillae contact with a posteromedial bone fragment which is probably formed by the palatine. As preserved, the right maxilla is 49.8 cm long. In ventral view, it widens in its posterior region. In lateral and dorsal view, the bone surface is smooth with only pitting ornamentation on its anterior half (Fig. 6). Shallow grooves extend a few millimeters posterior to some of the foramina but are barely visible and can only be spotted on the 3D model (see Material S2). Most other members of the Geosaurini subclade with known maxillae present conspicuous surface bone ornamentation made of grooves and ridges, often with a unique pattern (Young et al., 2013b). Plesiosuchus manselii and To. coryphaeus both have maxillae ornamented with grooves and raised ridges, while Dakosaurus maximus has additional pits to this ornamentation pattern (Young et al., 2012a, 2013b). The incomplete maxilla of cf. Torvoneustes (MANCH L6459) shows a similar pattern to To. coryphaeus with moderate to strong grooves and a raised edge aligned with the sagittal axis of the skull but lack posteroventral foramina (Young, 2014). Torvoneustes carpenteri has a pattern of pits and grooves on the lateral edge of its maxilla (Grange & Benton, 1996; Wilkinson, Young & Benton, 2008). In contrast, MJSN BSY008-465 has an ornamentation pattern closer to the one seen in Geosaurus giganteus, or Purranisaurus potens (Young & Andrade, 2009; Young et al., 2013b; Herrera, Gasparini & Fernández, 2015).

Figure 6 MJSN BSY008-465, holotype of Torvoneustes jurensis (Kimmeridgian, Porrentruy, Switzerland).

Photographs and scientific drawings of the right maxilla in (A, B) ventral and (C, D) lateral views. Anterior is to the right. Abbreviations: al, alveolus; for, foramen; mx, maxilla; no, notch; pa, palatine. Numbers indicate the preserved alveoli. Matrix is in yellow.

The alveoli are set on a slightly more dorsal plane than the palatal surface of the maxilla. The medial margin of the palatal plate of the right maxilla bears a longitudinal furrow that likely corresponds to the maxillary palatal groove. Each maxilla preserves at least 15 alveoli, which is more than the estimated number of 14 (with 11 strictly preserved alveoli) for To. carpenteri (Grange & Benton, 1996; Wilkinson, Young & Benton, 2008; Young et al., 2013b) and less than the estimated alveolar count of 17–19 for To. coryphaeus (Young et al., 2013b). However, there is at least one or two missing alveoli on MJSN BSY008-465 considering the dentary alveolar count. None of the specimens referred to the genus Torvoneustes preserves a complete maxilla, therefore this character should be treated carefully (see Discussion). Among the derived Geosaurini, maxillary alveolar count is usually estimated to be lower than 16 (Table 1; Wilkinson, Young & Benton, 2008; Young & Andrade, 2009; Young et al., 2012b). The alveoli are large, overall homogeneous in size, slightly anteriorly oriented and subcircular with a slightly longer anteroposterior axis than their mediolateral axis, which differentiates MJSN BSY008-465 from the members of the informal ‘E-clade’ taxa whose members show the opposite condition with alveoli wider than long (Abel, Sachs & Young, 2020). The interalveolar space is homogeneously narrow, being less than a quarter the length of the adjacent alveoli. Tooth enlargement and interalveolar space reduction are characteristics shared among several members of the Geosaurini subclade (Herrera et al., 2015) like Dakosaurus, Plesiosuchus (Young et al., 2012b), P. potens (Herrera, Gasparini & Fernández, 2015) some members of the ‘E-clade’ (Abel, Sachs & Young, 2020) and Torvoneustes (Grange & Benton, 1996; Wilkinson, Young & Benton, 2008; Young et al., 2013b; Young, 2014). The maxillary sutural surfaces with the nasal and premaxilla are visible in lateral and dorsal views. In medial view, the maxilla is concave to accommodate the nasal cavity and thicker around the tooth row. A notch between the 5th and 6th alveoli (Fig. 6A) on the right maxilla is a reception pit but is the only one on the fossil and therefore does not support a tooth-on-tooth vertical interlocking as seen in Dakosaurus (Young et al., 2012a) or Tyrannoneustes lythrodectikos (Foffa & Young, 2014). In addition, there is no evidence for a maxillary overbite as seen in Geosaurus giganteus (Young & Andrade, 2009). It is likely that the tooth occlusion of MJSN BSY008-465 follows the same interdigitated pattern as in To. mexicanus (Barrientos-Lara et al., 2016).

Table 1 Overview of the Kimmeridgian-Lower Tithonian metriorhynchids species and a few of their dental characteristics.

		Species	Age	Country	Teeth ornamentation	Carinae	Denticules	Ziphodonty	Denticules density (number of denticle/5 mm)	Maxillary tooth count (absolute)	Maxillary tooth count (estimated)	Dentary tooth count (absolute)	Dentary tooth count (estimated)	Specimen description	
Metriorhynchinae	Rhacheosaurini	Cricosaurus suevicus (Fraas, 1901, 1902, SMNS 9808)	Upper Kimmeridgian	Germany	Smooth	Yes, faint	No	/	/	26		24		Complete specimen in limestone	
Cricosaurus albersdoerferi (Sachs et al., 2021, BMMS-BK 1-2)	Upper Kimmeridgian	Germany	Smooth	Yes, faint	No	/	/	23	23+	22	22+	Complete specimen in limestone	
Cricosaurus elegans (Wagner, 1852, BSPG AS I
504)	Lower Tithonian	Germany	Smooth	Yes, faint	No							Skull in limestone	
Cricosaurus bambergensis (Sachs et al., 2019, NKMB-P-Watt14/274)	Upper Kimmeridgian	Germany	Smooth	Yes, faint	No	/	/	23	23+	18+	18+	Complete specimen in limestone	
Cricosaurus rauhuti (Herrera, Aiglstorfer & Bronzati, 2021, SNSB-BSPG 1973 I 195)	Lower Tithonian	Germany	Well-spaced, low longitudinal ridges	Yes, at least unicarenate	No	/	/	/	/	/	/	Incomplete skull	
Maledictosuchus nuyiviianan (Barrientos-Lara, Alvarado-Ortega & Fernández, 2018, IGM 4863)	Kimmeridgian	Mexico	Smooth labial side. Discontinuous low apicobasal ridges on the lingual surface	Yes	No	/	/	17	26?	/	/	Incomplete skull	
‘Cricosaurus’ saltillensis (Buchy, Young & Andrade, 2013, MUDE CPC 487)	Lower Tithonian	Mexico	Faint apicobasally aligned subparallel
ridges	Yes, faint	No	/	/	12	17	15	~15?	Disarticulated skull	
	Metriorhynchus palpebrosus (Phillips, 1871; Grange & Benton, 1996, (OUMNH
J.29823)	Lower Tithonian	United Kingdom	?	?	?	?	?	25 (OUMNH J.29823)-27	/	14	14+	Skull. No teeth remaining	
Metriorhynchus geoffroyii (Metriorhynchus brevirostris) (Young et al., 2020a; MHNG V-2232)	Lower Kimmeridgian	France	?	?	?	?	?	14	20+? (half of the rostrum is easily missing)	/	/	Anterior half of the rostrum, no teeth remaining	
Geosaurinae	Geosaurini	‘Metriorhynchus’ cf hastifer (Chouquet cf ‘hastifer’, Lepage et al., 2008; Eudes-Deslongchamps, 1867–1869)	Kimmeridgian	France	Conspicuous apicobasal ridges	Yes	?	?		20?	20+?	/	/	complete skull	
Dakosaurus maximus (Young et al., 2012b, SMNS 8203)	Upper Kimmeridgian	Germany	Overall smooth	Yes, prominent	Yes	Macro	16–18	13		12		Incomplete skull	
Plesiosuchus manselii (Young et al., 2012a, NHMUK PV OR40103)	Upper Kimmeridgian	United Kingdom	Low relief apicobasal ridges	Yes, prominent	Yes	Micro	?	14	14 to 18	13		Incomplete skull	
Geosaurus giganteus (Young & Andrade, 2009, NHM R.1229, NHM 37020)	Lower Tithonian	Germany	Overall smooth	Yes, prominent	Yes	Micro	?	12 (NHM 37020)	12+	7 (NHM 37020)	7+	NHM R.1229 : middle portion of the skull and mandible, deformed. NHM 37020 : skull and mandible in limestone.	
Geosaurus grandis (Young et al., 2012a, BSPG AS-VI-1)	Lower Tithonian	Germany	Smooth	Yes, prominent	Yes	Micro	28,1	14					
Torvoneustes carpenteri (Grange & Benton, 1996, BRSMG Ce17365)	Upper Kimmeridgian	United Kingdom	Conspicuous apicobasal ridges	Yes, prominent	Yes	Micro	?	11	14	/	/	Heavily crushed, incomplete skull	
Torvoneustes coryphaeus (Young et al., 2013b, MJML K1863)	Lower Kimmeridgian	United Kingdom	Conspicuous apicobasal ridges	Yes, prominent	Yes	Micro	?	11	17-19	/	/	Incomplete skull, half of the rostrum missing	
Torvoneustes mexicanus (Barrientos-Lara et al., 2016, IGM 9026)	Kimmeridgian	Mexico	Conspicuous apicobasal ridges	Yes, prominent	Yes	Micro	30	5		4		Fragmentary rostrum	
MJSN BSY008-465	Upper Kimmeridgian	Switzerland	Conspicuous apicobasal ridges	Yes, prominent	Yes	Micro	30–40	15	Up to 21	16	Up to 17	Disarticulated skull	
Note:

Gracilineustes acutus is excluded from the table due to the lack of information, the specimen was lost during WW2. The same goes for Rhacheosaurus gracilis NHMUK PV R 3948 for whom the teeth or alveoli are indistinguishable, while Rhacheosaurus cf gracilis LF 2426 skull is not entirely preserved. Therein the distinction between “true ziphodont” and “false ziphodont” condition as defined by Young et al. (2010) is not specified. The estimated tooth count is from the referred articles; the denticle density for D. maximus and Geo. grandis from Andrade et al. (2010) and for To. mexicanus from Barrientos-Lara et al. (2016).

Nasal

Both nasals are present as symmetrical, unfused elements (Figs. 5A, 5B). The right nasal is the best preserved, most complete and less deformed of the two. The nasals are flattened by taphonomy, the left one more so than the right one. The anterior process of the right nasal is curved upward due to postmortem deformation. The posterior part of the right nasal has been fractured then glued back but is not perfectly aligned. This same part is lost in the left nasal. The anterior part of the left nasal is broken and bent laterally. The bone surface is smooth. Only a few elliptical pits are present along the anterolateral parts (Figs. 5A, 5B). This ornamentation pattern differs from the one seen in To. coryphaeus (Young et al., 2013b), Pl. manselii, D. maximus (Young et al., 2012b), and cf. Torvoneustes (MANCH L6459) (Young, 2014). However, it is consistent with other species of Geosaurini with smooth and pitted nasals such as To. carpenteri (Grange & Benton, 1996), G. grandis, G. giganteus (Young & Andrade, 2009), and Dakosaurus andiniensis (Pol & Gasparini, 2009). The end of the dorsoposterior process, which should contact the lacrimal, is lost in both nasals. Overall, the nasals are triangular in shape, with the anterior part being elongate and acute. This shape is found in most metriorhynchoids (Andrews, 1913; Lepage et al., 2008) including rhacheosaurines such as Cricosaurus Wagner, 1858 (Fraas, 1902; Herrera, Fernández & Gasparini, 2013; Parrilla-Bel & Canudo, 2015; Sachs et al., 2019, 2021), derived geosaurins like Dakosaurus, Plesiosuchus, Torvoneustes, Purranisaurus (Wilkinson, Young & Benton, 2008; Pol & Gasparini, 2009; Young et al., 2012b, 2013b; Herrera, Gasparini & Fernández, 2015) and basal metriorhynchoid such as Pelagosaurus typus Bronn, 1841 (Pierce & Benton, 2006). Based on the skull reconstruction (Fig. 4), the nasals do not reach the premaxillae anteriorly. The posteromedial, posterolateral, and anterolateral sutural surfaces of the nasal with the frontal, the prefrontal, and the maxilla, respectively, are well preserved. On the median edge, a subtle angle marks the limit between the nasal-frontal and the median nasal sutures. The two nasals would contact on the midline of the skull, the medial concavity of the nasals in dorsal aspect suggests the presence of the longitudinal depression at their contact, a metriorhynchoid apomorphy (Young et al., 2012b).

Prefrontal

Both prefrontals are preserved, but the anterior parts and the descending processes are missing. The left prefrontal is still connected with the frontal (Fig. 7) while the right one is found apart (Fig. 5C). The right prefrontal seemingly suffered more damage than the left one, with the bone surface being partially flaked off and the bone flattened and stretched. It has also been broken and glued back together. The left prefrontal shows little deformation but is raised in its anterior part above the level of the frontal during taphonomy. The posterior part of the descending process on both prefrontals is barely preserved as a thin, concave structure. In dorsal view, the prefrontal is large, laterally extended, and it partially overhangs the orbit (Fig. 4A), a feature common within Metriorhynchidae (Fraas, 1902; Andrews, 1913; Herrera, Gasparini & Fernández, 2015; Young et al., 2010). The bone surface is smooth with numerous round or elliptical pits, as in P. potens (Herrera, Gasparini & Fernández, 2015), more densely distributed on the anteromedial part (Figs. 5C, 7). As in most metriorhynchids, the posterolateral corner of the prefrontal is rounded in MJSN BSY008-465 (Andrews, 1913; Lepage et al., 2008), which differs from the geosaurines To. coryphaeus (Young et al., 2013b) and D. maximus (Young et al., 2012a; Pol & Gasparini, 2009) in which the posterolateral corner is angular. The inflection point of this corner is directed posteriorly and forms with the midline of the skull an angle of about 70°, which is found in other Geosaurinae such as To. carpenteri, D. maximus and D. andiniensis. A posteriorly directed inflection point of the prefrontal of 70° or less is an apomorphy of the tribe Geosaurini (Cau & Fanti, 2011). If they share a similar shape, the posterior edge of the prefrontals in MJSN BSY0008-465 lacks the “finger-like” projections described in To. carpenteri (Young et al., 2013b).

Figure 7 MJSN BSY008-465, holotype of Torvoneustes jurensis (Kimmeridgian, Porrentruy, Switzerland).

(A) Scientific drawing, (B) interpretative drawing and (C) photograph of the frontal and left prefrontal in dorsal view. The angle formed by the lateral contacts of the frontal with the prefrontal and nasal is indicated in (B). Abbreviation: fr, frontal; if, intertemporal flange; prf, prefrontal. Matrix is in yellow.

Frontal

The frontal is mostly complete but somewhat flattened during taphonomy (Fig. 7). It consists of a single unpaired element without external sign of a medial suture. Only the anterior most part of the anteromedial process and portions of both intratemporal flanges are missing. The medioventral part of the left lateral process is extended in posteromedial direction (Fig. 7). Remaining matrix prevents the observation of the ventral side of the frontal. The anteromedial process of the frontal is broad, triangular in shape and extends anteriorly between the nasals. Considering the nasals geometry and preserved sutural contacts (Fig. 4), it is likely that the anterior process of the frontal ended in an acute tip, almost reaching the level of the anterior margin of the prefrontals as seen in cf. Torvoneustes (MANCH L6459) and To. coryphaeus (Young et al., 2013b; Young, 2014). This differs from the shorter process of To. carpenteri (Grange & Benton, 1996; Wilkinson, Young & Benton, 2008), but the crushing of the specimen might prevent an unbiased assessment of this character. Posteriorly, the frontal has two posterolateral processes forming the anterior margin of each supratemporal fenestrae, and a posteromedial process forming the anterior part of the intertemporal bar. The opening between the posterior and the lateral processes form an angle of approximately 60°, which is commonly found among geosaurines whereas metriorhynchines typically show an angle of about 90° (Wilkinson, Young & Benton, 2008). The posterior process is straight with a constant width of 23 mm and contact the parietal, forming a M-shaped, strongly digitated suture (Fig. 7). The intratemporal flanges (sensu Buchy, 2008) extend ventrally from the posterior and lateral processes. They form a triangular area with faint pitting inside the anterior corner of the supratemporal fenestrae. The anterior outline of the supratemporal fossa suggests that it was longer than wide and ovoid in shape, which is a common feature among geosaurines (Buchy, 2008). The supratemporal fenestrae extend far anteriorly and almost reach the level of the interorbital minimal distance (Fig. 7). This condition is seen in both To. carpenteri and To. coryphaeus as well as within G. grandis (Foffa & Young, 2014, fig. 9). In dorsal view, the external surface of the frontal is smooth with occasional pitting on the center of the bone, mostly concentrated on the anterior part of the posterior process. This differs from the ornamented frontal of To. coryphaeus (Young et al., 2013b), as well as from the anteroposteriorly aligned grooves and ridges observed in the frontal of cf. Torvoneustes (MANCH L6459; Young, 2014). In contrast, it resembles the description of To. carpenteri (Grange & Benton, 1996; Wilkinson, Young & Benton, 2008). In To. carpenteri and To. coryphaeus, the anterior and lateral processes of the frontal form an almost straight line or a slight concavity (Young et al., 2013b), whereas in MJSN BSY008-465 there is a clear angle of 142° between the processes, measured on the better preserved left side (Fig. 7B). The inflection point is probably located at the meeting point between the frontal, nasal and prefrontal.

Postorbital

Only the part forming the anterolateral margin of the supratemporal fossa of the right postorbital is preserved (Figs. 8A–8C). It is broken around its middle and was glued back together. The ventrolateral part, which should make a major contribution to the postorbital bar, is missing. The posterior portion of the postorbital curves slightly laterally but this is probably the result of postmortem deformation. Two processes are distinguishable on what is preserved of the postorbital. The anterior process is anteromedially oriented in dorsal view, making most of the curve of the lateral edge of the supratemporal fossa, and was in contact with the posterolateral process of the frontal. The posterior process is almost straight, slightly posteromedially oriented, and was in contact with the squamosal. The contact with the lateral process of the frontal is V-shaped, as described in several other metriorhynchids like Ty. lithrodectikos (Andrews, 1913; Foffa & Young, 2014). In the posterior part, the postorbital becomes a thin raised ridge that forms the anterior part of the postorbital-squamosal ridge. Posteriorly, the postorbital-squamosal suture is not readily visible but a change in the bone texture on the medial surface of the bone could match with a similar change on the anterior part of the right squamosal. In medial view, a deep incision in the postorbital marks the contact with the frontal (Fig. 8B), which agrees with observations made on complete skulls of other Metriorhynchidae (Andrews, 1913; Young et al., 2013b). This contact marks the point where the postorbital extends medioventrally to take part in the intratemporal flange. Based on the skull reconstruction, the postorbital likely extended further laterally than the prefrontal, resulting in an enlarged supratemporal fossa as usually seen in metriorhynchids (Wilkinson, Young & Benton, 2008; Foffa & Young, 2014). The bone surface is unornamented except for one elliptical foramen on the anterior process. There might be another foramen at the corner of the “V-shaped” suture but it is hard to discern due to poor bone preservation.

Figure 8 MJSN BSY008-465, holotype of Torvoneustes jurensis (Kimmeridgian, Porrentruy, Switzerland).

Posterior cranial elements with the right postorbital in (A) dorsal, (B) medial, and (C) lateral views; the parietal in (D) right lateral and (E) dorsal view; and the right squamosal in (F) dorsal view. Anterior to the right (except in B to the left). Abbreviations: in, incision; sqfs, squamosal flat surface.

Squamosal

Both squamosals are present but only their dorsal portion is preserved. The descending process, which participated to the occipital region, is lost. The right squamosal is the better preserved than the left. In dorsal view, the medial part of the right squamosal is concave (Fig. 8F), forming a characteristic “L-shape” (Andrews, 1913; Foffa & Young, 2014; Pol & Gasparini, 2009). The anterior part was in contact with the postorbital. This suture is not readily visible, but a change in bone texture could match with the one seen on the postorbital (see above). A marked ridge separates the squamosal in a medial and a lateral half. This ridge forms the posterior part of the postorbital-squamosal ridge, whose postorbital-squamosal contact would be an area of muscle attachment (Holliday & Witmer, 2007, 2009; Young et al., 2013b). This ridge is present in all Metriorhynchidae but significantly lower among Metriorhynchinae compared to Geosaurinae (Andrews, 1913; Lepage et al., 2008; Pol & Gasparini, 2009; Young et al., 2012b, 2013b; Young & Andrade, 2009). The median process of the squamosal forms the posterolateral corner of the supratemporal fossa and was probably in contact with the lateral process of the parietal, this contact however is not preserved. The posterior edge of the lateral process curves slightly upwards. The lateral part if the squamosal is a vertical descending surface forming the squamosal flat surface. This structure has been described in many metriorhynchids (Pol & Gasparini, 2009; Herrera, Fernández & Gasparini, 2013; Herrera, Gasparini & Fernández, 2015; Parrilla-Bel et al., 2013) and is not as well expressed as the one described for To. coryphaeus (Young et al., 2013b) but does resemble the one seen in D. andiniensis and Cricosaurus araucaniensis (Pol & Gasparini, 2009). There is a foramen in the middle of the squamosal on both medial and lateral sides. In ventral view, the squamosal presents several concavities, identical on both elements that may correspond to portions of the cranioquadrate canal and otic aperture, but they cannot be identified with certainty due to poor preservation.

Parietal

The parietal is a single unpaired element, as usual in crocodylomorphs (Figs. 8D, 8E; Pol & Gasparini, 2009; Leardi, Pol & Clark, 2017). The medial part of the parietal is well preserved, whereas its lateral processes are lost. Like the squamosal, only the dorsal part is preserved. The parts that should connect ventrally with the prootic and the laterosphenoid are missing. Upon discovery, the parietal and frontal were still articulated with one another (Fig. 2), but this contact was lost during extraction and preparation. Anteriorly, the parietal is of equal width with the posterior process of the frontal. It then gradually narrows posteriorly to an extreme degree until it only appears as a raised sagittal ridge (Figs. 4, 8E). This narrowing is more pronounced in MJSN BSY008-465 than in To. coryphaeus (Young et al., 2013b) and is at least as strong as in To. carpenteri (Grange & Benton, 1996), if not more. The anterior process of the parietal forms the posterior part of the intertemporal bar. Ventrally, the parietal widens to form the posteromedial corner of the supratemporal fossae. When complete, the parietal would be “T-shaped” in dorsal view with widely extended lateral processes as in other metriorhynchids (Andrews, 1913; Young et al., 2013b; Le Mort et al., 2022). In lateral view, the anterior process of the parietal slopes down posteriorly, starting from the point where the ridge is thinnest. From this point, the ridge widens again and forms a triangular area facing dorsally as in other Metriorhynchidae (Herrera, Gasparini & Fernández, 2015). The contact with the supraoccipital is not preserved. In ventral view, the posterior part of the parietal is hollow and has one or two foramina on its deepest point.

Mandible

Both mandibular rami are preserved (Fig. 4B). They are relatively complete, but some parts are fractured, eroded, and deformed. Some elements of the mandible are not articulated anymore. The mandible is represented by several main parts: two ensembles made of the angular, surangular, articular and prearticular; the disarticulated splenials; and the dentaries (preserved in several pieces). The coronoids are lost. The notable absence of a mandibular fenestra is an apomorphy of the Metriorhynchidae (Fraas, 1902; Andrews, 1913; Vignaud, 1995; Young et al., 2010). The total length of the mandible is about 815 mm with the anterior end missing. The general shape of the mandible is similar to the one described for Ty. lythrodectikos and Geosaurini with the coronoid process located higher than the plan of the tooth row and lower than the retroarticular process, indicating an increase in gape (Young et al., 2012a, 2012b, 2013a, Foffa & Young, 2014).

Dentary

Both dentaries are preserved but not equally well. The left dentary is almost complete but broken into three parts (Fig. 9), only missing its anteriormost portion. The right dentary is broken into two pieces but is highly damaged and deformed. Its ventral and medial parts are completely lost, whereas the posterior part is crushed and deformed. The latter now lies more dorsally and medially compared to the anterior part. The damages are partly due to taphonomic conditions, but also to the poorly controlled acid treatment of the fossil. The left dentary is about 45 cm long, 4.5 cm high and 2.6 cm wide. The right dentary preserves two teeth, including one on its most deformed part with the alveolus deformed and projected inward (Fig. 4B). The left dentary preserves one tooth in the middle of the rostrum (Fig. 9). In dorsal view, the dentary is a long and thick bone. It narrows posteriorly where it was in contact with the angular and surangular. The dentary widens anteriorly starting from the posteriormost alveolus, which is smaller than the one immediately in front. The anterior part of the dentary, including the three anteriormost visible alveoli, shortly curves inward, this bending of the dentary seems natural. There are at least 16 clearly identifiable alveoli, potentially 17, that are overall homogeneous in size. The alveoli are enlarged, subcircular, and slightly longer than wide. The three anteriormost alveoli are larger than the others and lie slightly more dorsally. Overall, the more anterior are the alveoli the more anterodorsally they are oriented. The remaining tooth on the left dentary is anterodorsally directed while the one on the undeformed part of the right dentary is dorsally oriented. As in the maxillae, the interalveolar space is uniform and greatly reduced, being less than half of the anteroposterior length of the adjacent alveoli (see above). In lateral view, the lateral margin of the alveoli lies slightly lower than the medial margin except for the three anteriormost alveoli where the lateral and medial margins are on the same plane (Figs. 9B, 9D). The surangular-dentary groove is visible (Fig. 9B), and does not end anteriorly with an enlarged foramen as it does in Dakosaurus (Pol & Gasparini, 2009). A pitting pattern is present on the lateral and ventral surfaces of the dentaries. However, there is no heavy grooving as described in To. carpenteri (Wilkinson, Young & Benton, 2008). At least eight alveoli are adjacent to the symphysis. This count is higher than the four observed in Dakosaurus maximus (Young et al., 2012b) but close to estimated height in the indeterminate Geosaurini (SMNS 80149) from the informal E-clade (Abel, Sachs & Young, 2020; Young et al., 2020a) and Geosaurus (Young et al., 2012b); as well as the nine of Pl. manselii (Young et al., 2012b), but lower than the 10–13 of the early Geosaurini Ty. lythrodectikos (Young et al., 2013a; Waskow, Grzegorczyk & Sander, 2018). No reception pits for maxillary teeth are observed on the dentaries, indicating that there was no overbite creating a scissor-like occlusion mechanism like in G. giganteus nor any tooth-on-tooth interlocking as seen in D. maximus (Young et al., 2012a; Young & Andrade, 2009).

Figure 9 MJSN BSY008-465, holotype of Torvoneustes jurensis (Kimmeridgian, Porrentruy, Switzerland).

Photographs of the left dentary in (A) dorsal and (B) lateral views. Drawings of the left dentary in (C) dorsal and (D) lateral views. Anterior is to the left. Abbreviations: al, alveolus; for, foramen; sdg, surangulodentary groove; tc, tooth crown. Matrix is in yellow.

Splenial

The left splenial is better preserved than the right one. The two splenials have been flattened and both their anterior and posterior ends are broken off. The left splenial is about 37 cm long. In dorsal view, each splenial is narrow on its anterior and posterior ends and thicker in the middle. The lateral edge is straight, but the medial edge is slightly convex (Fig. 10C). The lateral aspect of the splenial is overall concave, with raised ridges forming the sutural contacts with the angular and dentary ventrally, and with the dentary and surangular dorsally (Fig. 10A). Anteriorly, the splenial likely reaches the fifth or sixth alveoli, but this observation might be obscured by the poor preservation of the specimen in this area. In Pl. manselii (Young et al., 2012b), the splenial anteriorly reaches alveoli six anteriorly and alveoli seven or eight in Ty. lythrodectikos (Foffa & Young, 2014; Waskow, Grzegorczyk & Sander, 2018) and in Rhacheosaurini such as Cricosaurus bambergensis or C. albersdoerferi it is clearly posterior to the 10th alveoli (Sachs et al., 2019, 2021; L. Girard, 2021, personal observation). The medial surface of the splenial is slightly convex. The ventral edge of the bone is thicker and participates to the ventral edge of the mandible itself. The bone surface is smooth. On the medial surface, a foramen is visible posterior to the thickened part of the bone. The splenials would contact one another along the symphysis and the remains of the contact is visible on the anterior third of the splenial, marked by a rougher surface of the bone (Figs. 10B, 10D).

Figure 10 MJSN BSY008-465, holotype of Torvoneustes jurensis (Kimmeridgian, Porrentruy, Switzerland).

Photographs of the right splenial in (A) lateral, (B) medial, and (C) dorsal views. Photograph of the left splenial in (D) medial view. Abbreviations: for, foramen; r, ridge; sym, symphysis. Arrows point anteriorly and dashed lines show limits of the symphysis marks.

Angular and surangular

The angular and surangular are preserved on both sides. They are missing their anterior part and show signs of crushing, erosion, and deformation, especially on the left mandibular ramus. The angular and surangular are strongly sutured along their entire length. They respectively form the ventral and dorsal halves of the posterior part of the mandibular ramus (Fig. 11). In lateral view, the surangular-dentary groove is well expressed. This groove is present in all Metriorhynchidae, albeit not always visible due to deformation, and it is especially deep in the members of the Geosaurini tribe (Pol & Gasparini, 2009; Young & Andrade, 2009; Young et al., 2012b, 2013b). This groove is associated with the passage of the mandibular nerve (Holliday & Witmer, 2007; Young & Andrade, 2009; George & Holliday, 2013) and its posterior end is marked by a foramen. Two other smaller foramina, more or less aligned with the one on the surangular-dentary groove, are found on the posterior part of the surangular, following the upward curve of the bone. The dorsal margin of the surangular rises slightly posteriorly before sloping down after reaching the coronoid process. The coronoid process is located on the anterodorsal half of the surangular. The coronoid process is higher than the tooth row, but lower than the retroarticular process. In dorsal view, the coronoid process narrows posteriorly and forms a ridge. In medial view, the surangular medial ridge for the coronoid is visible. The medial surface of the angular and surangular is concave, especially on their anterior part where they would form the lateral wall of the Meckelian groove and contact the splenial and the coronoid. The large foramen that connects with the surangular-dentary groove in the lateral surface of the ramus is present below the coronoid process. The angular is thicker than the surangular and forms the ventral margin of the mandible. The posterior part of the angular curves upward towards the retroarticular process at an angle of approximately 30° with the ventral surface of the angular. Posteriorly, the angular extends beyond and rises higher than the glenoid fossa to form the ventral part of the retroarticular process as on other Geosaurini (Young et al., 2012b; Herrera et al., 2015).

Figure 11 MJSN BSY008-465, holotype of Torvoneustes jurensis (Kimmeridgian, Porrentruy, Switzerland).

Photographs of the posterior part of the right ramus of the mandible in (A) medial, (B) dorsal, and (C) lateral views. Scientific drawings of the posterior part of the right ramus of the mandible in (D) medial, (E) dorsal, and (F) lateral views. Abbreviations: an, angular; art, articular; cp, coronoid process; for, foramen; gf, glenoid fossa; pra, prearticular; ret, retroarticular process; san, surangular; sdg, surangulodentary groove. Matrix is in yellow.

Prearticular

Only the right prearticular is well preserved. The prearticular is absent in many crocodylomorphs (Iordansky, 1973; Ruebenstahl et al., 2022) but is present in metriorhynchoids as well as in a few teleosaurids such as Machimosaurus (Andrews, 1913; Martin & Vincent, 2013; Young et al., 2014a; Martin, Vincent & Falconnet, 2015) Its presence is therefore considered a symplesiomorphy of Thalattosuchia (Andrews, 1913; Martin, Vincent & Falconnet, 2015). The prearticular is a triangular-shaped bone only visible in medial view of the ramus (Figs. 11A, 11B). It contacts the angular ventrally, the surangular laterally, and the articular posterodorsally.

Articular

The articular is well preserved only on the right mandibular ramus. It contacts the prearticular anteroventrally, the angular posteroventrally, and the surangular laterally. The articular projects far medially. The glenoid fossa, which accommodates the articular condyle of the quadrate, is deep and rounded, and is oriented anterodorsally. The glenoid fossa is divided in two concavities by a low ridge similar to the condition in To. coryphaeus and Ty. lithrodectikos (Young et al., 2013b; Foffa & Young, 2014) but different to what is seen in Pl. manselii where there is no separation (Young et al., 2012b). The glenoid fossa is separated from the retroarticular process by a raised ridge similar to the one seen in To. coryphaeus, Ty lithrodectikos or “Metriorhynchus” brachyrhynchus which forms “ridge-and-concavity” morphology to accommodate the “sulcus-and-condyle” of the quadrate (Andrews, 1913; Young et al., 2012b, 2013b; Foffa & Young, 2014). The dorsal surface of the retroarticular process is a triangular smooth posterodorsally oriented concavity, which curves medially in dorsal view. The retroarticular process extends laterally beyond the glenoid fossa and slopes downward in medial view. The medial end of the bone is a rugose surface. This overall matches the usual metriorhynchid condition (Andrews, 1913; Young et al., 2012b, 2013b; Foffa & Young, 2014). The tip of the retroarticular process is broken off, but it would rise higher than the rest of the articular.

Dentition

At least fifteen isolated teeth were found closely associated with the skeleton MJSN BSY008-465. Nine of these isolated teeth are complete, or almost complete, and preserve both the crown and the root. Three additional teeth are still in place on the dentaries and two more on the left maxilla (see above).

Teeth

Based on macroscopic observation, the teeth correspond to the typical metriorhynchid morphology. They are caniniform, conical, and single-cusped (Massare, 1987; Vignaud, 1997). The teeth are large, robust and bicarinate with the carinae on the anteroposterior axis running continuously from the crown base to the apex. There is no basal constriction of the crown, but the crown-root junction is clearly visible from color and texture (Fig. 12). The length of each tooth roots is at least twice the height of each crown. In cross section, the base of the crown is sub-circular to ovoid with the labial face thicker than the lingual one. The tooth roots are ovoid in cross-section. Closer to the apex, the mediolateral compression of the teeth is increasing. Most tooth crowns are over 20 mm high, but the tooth height is not uniform. The average crown height is 21.94 mm with a standard deviation of 4.38. These measures includes all of the 15 teeth. It is to be noted that some of them have their apex broken but not enough to significantly affect the crown height. The shortest tooth crown is 13.23 mm high and the highest is 29.28 mm. The average width at the tooth base is 10 mm (standard deviation 0.9) for a length of 11.36 in average (standard deviation 0.87). The average difference between the width and the length is 1.36 mm (standard deviation 0.93), but the minimal value is 0.3 mm and the maximal value is 3.67 mm with no correlation to the crown height. The teeth are curved lingually and posteriorly. The morphology of the alveoli and the teeth still in place, especially the anterior ones, indicate that teeth were implanted slightly forward in the jaws. Crowns are heavily ornamented with long longitudinal subparallel ridges on at least the basal two thirds of the crown. Ridges are denser on the lingual face than on the labial one. On the apex, the ornamentation becomes low, short ridges forming an anastomosed pattern of drop-shaped ornaments (Figs. 12, 13). This distinctive ornamentation pattern has only been described in other Torvoneustes species (Andrade et al., 2010; Young et al., 2013b; Barrientos-Lara et al., 2016; Foffa, Young & Brusatte, 2018; Young et al., 2019) as well as in the machimosaurids Machimosaurus von Meyer, 1837 and Lemmysuchus obtusidens Andrews, 1909 (Johnson et al., 2017). As seen in To. carpenteri, To. mexicanus and isolated Torvoneustes teeth (Andrade et al., 2010; Young et al., 2013b, 2014b, 2019; Barrientos-Lara et al., 2016; Foffa, Young & Brusatte, 2018; Madzia et al., 2021), the enamel ridges on the upper part of the teeth shift and bend toward the carinae. The carina is well developed, as in other Torvoneustes specimens. The developed keel and the shift of enamel ridges toward the carinae are both autapomorphies of Torvoneustes (Andrade et al., 2010; Young et al., 2013b, 2014b, 2019; Barrientos-Lara et al., 2016; Foffa, Young & Brusatte, 2018; Madzia et al., 2021). On the basal two thirds of the crown, the carinae are serrated with no involvement of the enamel ornamentation. On the apical third of the crown, there is a clear shift in the enamel ornamentation pattern, and the enamel ridges bend toward the carinae and touch the serrated keel (Fig. 13A). The serration is present from the base of the carinae to the apex and is high, especially on the apical half of the crown. Both of these features are also found in Torvoneustes (Andrade et al., 2010; Young et al., 2013b, 2014b, 2019; Barrientos-Lara et al., 2016; Foffa, Young & Brusatte, 2018; Madzia et al., 2021). The serration is only faintly visible on macroscopic observation. The tooth apex is sharp, similar to what is described for Torvoneustes mexicanus and different from the blunter tooth tips of To. carpenteri, To. coryphaeus and Torvoneustes teeth from the UK (Wilkinson, Young & Benton, 2008; Andrade et al., 2010; Young et al., 2013b, 2019; Barrientos-Lara et al., 2016; Foffa, Young & Brusatte, 2018; Madzia et al., 2021). In an unpublished work on isolated thalattosuchian teeth from the Pal A16 collection, Schaefer (2012) already noted the resemblance of the teeth of MJSN BSY008-465 with the ones of To. carpenteri, while pointing out the greater sharpness of the former.

Figure 12 MJSN BSY008-465, holotype of Torvoneustes jurensis (Kimmeridgian, Porrentruy, Switzerland).

Two of the best-preserved isolated teeth in (A) lateral and (B) probably anterior views. Abbreviations: c, crown; ca, carina; r, root.

Figure 13 MJSN BSY008-465, holotype of Torvoneustes jurensis (Kimmeridgian, Porrentruy, Switzerland).

Microscopic photographs of MJSN BSY008-465 teeth. (A, B) mid-upper tooth crown; (C) tooth apex; (D) tooth base. ib, inflated base; fd, false denticle and td, true denticle.

Observed with optic aids, the serration of the carinae is formed by a continuous row of poorly isomorphic, isolated denticles weakly affecting the keel height (poorly developed incipient denticles). This serration corresponds to the microziphodont condition as defined by Andrade et al. (2010) with the denticles all smaller than 300 µm in height and length. The denticles base length varies between 120 and 200 µm with an average of 160 µm (Fig. 13), differing from the regular and 142-µm-long (average) denticles of To. mexicanus (Barrientos-Lara et al., 2016; Table 1). The denticle density (number of denticles/5 mm, following Andrade et al., 2010) ranges from 31 to 40 depending on the area on the teeth with an average of 35, which is higher than the average measured on To. mexicanus (Barrientos-Lara et al., 2016) and G. grandis, but corresponds to the number found on Geosaurus indet. (SMNS 81834; Andrade et al., 2010; Table 1). The denticle density increases in the middle of the tooth crown and decreases at the apex, making the denticles more densely packed and narrower on the part where the carina is the highest. At the apex, the enamel ornamentation joins the carina (Figs. 13A, 13B), similar to the condition in To. carpenteri and To. mexicanus (Andrade et al., 2010; Barrientos-Lara et al., 2016) and contrary to what is observed in To. coryphaeus (Young et al., 2013b; Foffa et al., 2017). The denticles are harder to discern in this region. This false-ziphodont condition (Prasad & de Lapparent de Broin, 2002) is found in Torvoneustes but also in Machimosaurus (Andrade et al., 2010; Young et al., 2014b). However, the combination of true and false ziphodonty, as defined by Andrade et al. (2010), is only known in To. carpenteri, To. mexicanus, and Torvoneustes sp. (OUMNH J.50061 and OUMNH J.50079-J.50085; Young et al., 2013b, 2019; Barrientos-Lara et al., 2016). The base of the carina around the upper middle of the tooth shows structures resembling the “inflated base” seen in To. carpenteri (Fig. 13B; Andrade et al., 2010). MJSN BSY008-465 does not present the faceted teeth of Ieldraan and Geosaurus (Young et al., 2013a; Foffa et al., 2017), nor the macroscopic denticles of Dakosaurus (Andrade et al., 2010; Pol & Gasparini, 2009), nor the characteristic flanges on the side of the carinae seen in Plesiosuchus (Owen, 1883; Young et al., 2012b; Table 1).

Tooth count

The holotype of To. carpenteri (BRSMG Ce17365) is the only other specimen referred to Torvoneustes that has a maxilla as complete as MJSN BSY008-465 (Grange & Benton, 1996; Wilkinson, Young & Benton, 2008). The paratype of To. carpenteri (BRSMG Cd7203) preserves a dentary, yet just a few fragments and not a complete piece, in contrast to MJSN BSY008-465. On the right maxilla of MJSN BSY008-465, there are at least 15 preserved alveoli (13 on the left one) while there are at least 16 to 17 alveoli on the left dentary and possibly 17 alveoli on the more poorly preserved right dentary. The dentary tooth count is usually lower than the maxillary count. This might indicate at the very least one to two missing alveoli on the right maxilla. Metriorhynchids bear three teeth on the premaxilla (Andrews, 1913; Wilkinson, Young & Benton, 2008; Young & Andrade, 2009). Therefore, this would indicate a dental formula for MJSN BSY008-465 of three premaxillae teeth, 16 to 18 maxillary teeth and 16 to 17 dentary teeth (3+16−18/16−17) for a minimal count, but it is likely that the maxillary tooth count could be even higher. It is therefore closer to the 3+17−19/15−17 estimated for To. coryphaeus than the 3+14/14 formula estimated for To. carpenteri (Young et al., 2013b; Table 1). However, it must be noted that no Torvoneustes species was found with complete maxillae. Moreover, the skull of To. carpenteri (BRSMG Ce17365) is damaged, so the estimated tooth count might be underestimated, especially considering that, in To. coryphaeus, the maxillary tooth row reaches beyond the anterior margin of the orbit. This is also observed in other geosaurines such as D. andiniensis, G. giganteus and potentially Pl. manselii (Young et al., 2013b). Comparison with To. carpenteri and MJSN BSY008-465 are limited because the contour of the orbits is completely lost in both.

Tooth wear

In addition to postmortem fractures, the teeth of MJSN BSY008-465 present signs of macroscopic wear. Several teeth have their apex broken resulting in a flattened and smoothed tip. This type of wear was described in To. coryphaeus as an indication of repeated impact against hard surfaces. Some teeth also present signs of enamel spalling wear, mainly represented by triangular facets of broken enamel on the labial face. This type of wear was interpreted as resulting from tooth-food abrasion and was also described in To. coryphaeus, To. carpenteri and D. maximus (Grange & Benton, 1996; Andrade et al., 2010; Young et al., 2012a, 2012b, 2013b).

Postcranial elements

From the postcranial skeleton, many vertebrae and ribs are preserved as well as a few elements of the pelvis and hindlimbs. Despite the number of postcranial elements, no osteoderms were found. The absence of osteoderms is an apomorphy of Metriorhynchidae (Fraas, 1902; Andrews, 1913; Young et al., 2010).

Vertebrae

MJSN BSY008-465 was found with 22 of its vertebrae including three cervicals, nine dorsals and 10 caudals (Fig. 14). The atlas-axis complex is missing as well as the sacral vertebrae. The vertebrae suffered different level of damage and deformation. Some show stretching, with the centrum deflected from its natural, vertical plane and the apophyses not aligned anymore, or crushing with no preferential direction of deformation. This is mainly the case in the dorsal vertebrae. All vertebrae are amphicoelous, as in all metriorhynchids (Fraas, 1902; Pierce & Benton, 2006; Cau & Fanti, 2011; Young et al., 2013a; Parrilla-Bel & Canudo, 2015). The concavity is shallow and similarly developed in the anterior and posterior articular surfaces.

Figure 14 MJSN BSY008-465, holotype of Torvoneustes jurensis (Kimmeridgian, Porrentruy, Switzerland).

Cervical vertebra in (A) anterior and (B) right lateral views; dorsal vertebra in (C) anterior and (D) right lateral views; anterior caudal vertebra in (E) anterior and (F) right lateral views; Posterior caudal vertebrae in (G) anterior and (H) right lateral views. Abbreviations: di, diapophysis; pa, parapophysis; prz, prezygapophysis; pz, postzygapophysis.

Cervical vertebrae

Three post-axis cervical vertebrae are preserved. The number of post-axis cervical vertebrae is considered to be five among Metriorhynchidae, with the fifth cervical closely resembling the first dorsal (Fraas, 1902; Wilkinson, Young & Benton, 2008; Young et al., 2013a; Parrilla-Bel & Canudo, 2015; Sachs et al., 2021). The cervical vertebrae of To. jurensis look similar to one another, we can therefore assume that the preserved vertebrae are all mid cervicals. Two of them are well preserved, including one complete (Figs. 14A, 14B). The centrum is subcircular to ovoid, with the length of the vertebra subequal to the centrum height and width, which is typical for geosaurines (Parrilla-Bel & Canudo, 2015). The neural spine is shorter than the centrum height. As in all thalattosuchians, the cervical vertebrae are amphicoelous (Fraas, 1902; Wilkinson, Young & Benton, 2008; Pierce & Benton, 2006; Cau & Fanti, 2011; Young et al., 2013a; Parrilla-Bel & Canudo, 2015; Le Mort et al., 2022). The parapophysis is low on the centrum, ventrally directed, without reaching lower than the ventral margin of the centrum (Fig. 14). A low parapophysis not associated with the neural arch is what characterizes cervical vertebrae in metriorhynchids (Andrews, 1913; Young et al., 2013a; Parrilla-Bel & Canudo, 2015). The parapophysis ends with a concave articular facet. This facet articulates with the cervical rib. The diapophysis is also low, starting just above the neural arch-centrum suture, and ventrally oriented, reaching below the suture. The neurocentral sutures are not closed. On the ventral margin, at the edges of the articulation surfaces of the centrum, discrete ridges are a sign of muscle attachments (Parrilla-Bel & Canudo, 2015). There is a deep concavity between the parapophysis and diapophysis. This strong constriction has also been noted in Maledictosuchus riclaensis (Parrilla-Bel & Canudo, 2015). In ventral view, there are two shallow concavities between the parapophyseal processes and the centrum ventral margin, creating a medial keel also described on the cervical vertebrae of D. maximus and Ma. riclaensis (Fraas, 1902; Parrilla-Bel & Canudo, 2015). The ventral margin of the centrum is concave in lateral view. The parapophyses project below this margin in the middle of the centrum, but they do not extend more ventrally than the ventral margin of the articular facets. The zygapophyses are well developed, separated, and extended beyond the centrum. The postzygapophyses are wider than the prezygapophyses, but the latter extend further from the centrum. The articular surfaces of the zygapophyses are ovoid and flat. The morphology of the cervical vertebrae is consistent with the ones described for To. carpenteri, Ma. riclaensis and other metriorhynchids (Fraas, 1902; Andrews, 1913; Wilkinson, Young & Benton, 2008; Young et al., 2013a; Parrilla-Bel & Canudo, 2015; Le Mort et al., 2022).

Dorsal vertebrae

At least nine dorsal vertebrae are preserved, none of them with a complete neural spine nor complete diapophyseal processes. All of them are deformed to some extent. In metriorhynchids, the first dorsal vertebra is the one where the parapophysis is no longer on the centrum but on the neural arch (Young et al., 2013a; Parrilla-Bel & Canudo, 2015). The dorsal vertebrae of MJSN BSY008-465 follow the trend observed by Fraas (1902) with a constriction of the middle of the centrum giving it an hourglass shape. The centrum is higher than wide. Its length is subequal to its height. The vertebrae are amphicoelous, with both articular faces overall equally concave, unlike what had been noted in Ty. lythrodectikos and To. carpenteri (Wilkinson, Young & Benton, 2008; Young et al., 2013a; Parrilla-Bel & Canudo, 2015). On the neural arch, the spine extends vertically in the posterior half of the centrum length. The parapophysis joins the diapophysis on the neural arch to form the anterior extension, like a little step, of the transverse apophysis, as typically observed within Metriorhynchidae (Figs. 14C, 14D; Andrews, 1913, fig. 62; Parrilla-Bel & Canudo, 2015). The transverse apophysis extends greatly beyond the centrum on each side. It is overall straight with a slight ventral concavity. The zygapophyses are well developed, but not as much as the ones on the cervical vertebrae. As in Ty. lythrodectikos and the E-clade member PSHME PH1, they project slightly beyond the centrum (Parrilla-Bel & Canudo, 2015; Abel, Sachs & Young, 2020). Among the dorsal vertebrae of MJSN BSY008-465, some of them show unfused suture between the centrum and neural arch, indicating a specimen which did not achieved full maturity (a sub-adult; Brochu, 1996; Herrera, Fernández & Gasparini, 2013). As seen on the cervical vertebra, on the ventral margin of the caudal vertebra the edges of the articulation surfaces of the centrum, discrete ridges are a sign of muscle attachments.

Caudal vertebrae

In MJSN BSY008-465, only ten caudal vertebrae are preserved. The number of caudal vertebrae can differ greatly between metriorhynchid species but it is usually over 30 (Fraas, 1902; Andrews, 1913; Parrilla-Bel & Canudo, 2015; De Sousa Oliveira et al., 2023). The size and shape of the preserved vertebrae vary greatly, which suggests that they originate from different parts of the tail (Figs. 14E–14H). However, none of them can be associated with the bend of the tail fluke. Three vertebrae with reduced apophysis are associated with the anterior part of the tail (Figs. 14E, 14F). These caudal vertebrae are hourglass shaped, but the constriction is not as strong as in the cervical and dorsal vertebrae. Their centrum presents a ventral keel. The centrum length of the vertebrae is subequal to their width. The neural spine is overall rectangular in shape and oriented posterodorsally. The general shape of the caudal vertebra of MJSN BSY008-465 is similar to To. carpenteri and E-clade member PSHME PH1 (Wilkinson, Young & Benton, 2008; Abel, Sachs & Young, 2020). The other caudal vertebrae do not all preserve the neural spine. These vertebrae are greatly reduced for some of them, and are interpreted as more posterior in the caudal series. The zygapophyses are not preserved. The articular surfaces of the centra are rounded and slightly concave. On the caudal vertebrae, the suture between the centrum and neural arch is closed.

Ribs

Cervical ribs

Three cervical ribs are identified. They form short, slender ribs directed posteriorly with an acute end (Figs. 15A–15C). The external face forms a ridge starting from the tuberculum and capitulum. The medial face is concave. In cross section, the cervical ribs are roughly triangular at mid-shaft. In lateral view, the ribs are “V-shaped”. This correspond to the typical condition in Metriorhynchidae (Andrews, 1913; Wilkinson, Young & Benton, 2008; Abel, Sachs & Young, 2020).

Figure 15 MJSN BSY008-465, holotype of Torvoneustes jurensis (Kimmeridgian, Porrentruy, Switzerland).

(A–C) cervical ribs, (D, E) dorsal ribs; chevron in (F) dorsal and (G) lateral views (anterior to the top).

Dorsal ribs

There are 15 preserved, but incomplete, dorsal ribs from MJSN BSY008-465. The dorsal ribs are long and slender, ovoid or round in cross section (Figs. 15D, 15E). They are arched, having a flat medial surface, whereas the lateral surface is rounded. There is a ridge on the posterior surface running down from the tubercular (diapophyseal process). The proximal two thirds are flat on the medial side. In its distal third in lateral view, the rib narrows to form a ridge. On the medial side this part shows a sutural surface. This shift might mark the limit between the vertebrocostal part to the intercostal part of the rib, but no intercostal is sufficiently preserved to give a better description. Tuberculum and capitulum are not preserved in available ribs. The ribs overall resemble the dorsal ribs of To. carpenteri and other metriorhynchids (von Arthaber, 1906; Andrews, 1913; Wilkinson, Young & Benton, 2008; Herrera, Fernández & Gasparini, 2013, fig. 7N).

Chevron

At least one chevron is preserved, probably from the posterior part of the tail considering its small size (Figs. 15F, 15G). In lateral view it is Y-shaped, and in dorsal view it presents two lateral and one medial branches, which corresponds to what has been described in other Metriorhynchidae (Andrews, 1913; Sachs et al., 2019, 2021).

Appendicular skeleton

Ischium

The left ischium is badly preserved, lacking a major portion of its ventral part where the bone would widen the most (Fig. 16C). The proximal part is also missing, as well as the anterior process it should be bearing (Andrews, 1913; Wilkinson, Young & Benton, 2008; Herrera, Fernández & Gasparini, 2013; Young et al., 2013b). Like in other metriorhynchids, the neck of the ischium is narrow and thick, measuring 2.5 cm wide. The ischium widens and flattens distally in an overall triangular wing with a thickness of about two millimeters only. The partly preserved wing is similar to the classic metriorhynchid morphology as seen in To. carpenteri, C. araucanensis (Andrews, 1913; Wilkinson, Young & Benton, 2008; Herrera, Fernández & Gasparini, 2013; Le Mort et al., 2022). The surface of the wing is covered by striations corresponding to muscular attachment marks on both sides but better expressed on the lateral one.

Figure 16 MJSN BSY008-465, holotype of Torvoneustes jurensis (Kimmeridgian, Porrentruy, Switzerland).

Right femur in (A) medial and (B) lateral views; (C) left ischium in lateral view; (D) right fibula in lateral view.

Femur

The right femur of MJSN BSY008-465 lacks both proximal and distal ends (Figs. 16A, 16B). It has the sigmoidal shape typically found in Thalattosuchia (Andrews, 1913; Wilkinson, Young & Benton, 2008; Herrera, Fernández & Gasparini, 2013; Young et al., 2013b; Sachs et al., 2021) but with the curves not as pronounced as in Enaliosuchus macrospondylus Koken, 1883 or Ty. lythrodectikos (Young et al., 2013b; Sachs, Young & Hornung, 2020). The bone is about 24 cm long and 3 cm wide, a similar size to Ty. lythrodectikos (Young et al., 2013b). It is slightly narrower in the middle than at the ends. The medial side is almost flat, whereas the lateral side is convex. The bone flattens toward the distal end. As in Ty. lythrodectikos and C. araucanensis, the femur widens toward the distal end (Herrera, Fernández & Gasparini, 2013; Young et al., 2013b; Sachs et al., 2021). The proximal end, despite the damages it suffered, shows on both sides the rugose surface for muscle attachment commonly found in metriorhynchids (Andrews, 1913; Lepage et al., 2008; Wilkinson, Young & Benton, 2008; Herrera, Fernández & Gasparini, 2013; Young et al., 2013b).

Fibula

The right fibula is a slender bone about a third of the preserved femur length (Fig. 16D), the same proportion as found in Ty. lythrodectikos, C. suevicus and C. araucanensis (Andrews, 1913; Herrera, Fernández & Gasparini, 2013; Young et al., 2013b). This ratio is affected by the missing ends of the femur, as well as the missing part of the fibula. The hindlimb proportion in metriorhynchids are often measured with the femur and tibia. The tibia length being 30% to 40% of the femoral length is the general trend found among metriorhynchids while a ratio below 30% is distinctive of derived metriorhychins (Andrews, 1913; Wilkinson, Young & Benton, 2008; Young & Andrade, 2009; Cau & Fanti, 2011; Young et al., 2013b; Foffa et al., 2019). The distal end of the fibula is not preserved. The proximal end is damaged but shows a single convex condyle as usual in Metriorhynchidae (Andrews, 1913; Herrera, Fernández & Gasparini, 2013; Sachs et al., 2019, 2021). The inner side is flatter than the outer one, like what is seen in the femur. The bone is one centimeter wide at mid length and enlarged toward both ends, being 2 cm wide at the proximal end.

Phylogenetic analysis

Following the methodological protocol of Young et al. (2020b), the eight parsimony analyses resulted in eight strict consensus topologies with lengths ranking from 2,417 steps for the unweighted analysis to 2,033 steps for the weakly downweighted topologies (k = 20 and k = 50; see Table 2). The complete eight strict consensus trees are provided in Material S3, while descriptive statistics for each of these trees are presented in Table 2. Focusing on the internal relationships of Geosaurinae, four distinct topologies are recovered (Fig. 17). They all have a similar structure, except for the position of Tyrannoneustes lythrodectikos, Ieldraan melkshamensis, Geosaurus lapparenti, “Metriorhynchus” westermanni, and “Metriorhynchus” casamiquelai.

Table 2 Descriptive statistics of the cladograms resulting from the parsimony analysis.

K	Number of MPCs	Length	CI	RI	RC	HI	
–		2,417	0.334	0.803	0.268	0.666	
1	297	2,072	0.389	0.845	0.329	0.611	
3	297	2,065	0.391	0.846	0.331	0.609	
7	99	2,044	0.395	0.848	0.335	0.605	
10	99	2,044	0.395	0.848	0.335	0.605	
15	33	2,034	0.397	0.85	0.337	0.603	
20	33	2,033	0.397	0.85	0.337	0.603	
50	33	2,033	0.397	0.85	0.337	0.603	
Note:

In the unweighted analysis, the number of Most Parsimonious Cladograms is higher than the storage capacity (20,000).

Figure 17 Phylogenetic placement of MJSN BSY008-465 (Torvoneustes jurensis sp. nov.) within Geosaurinae in the parsimony analyses.

(A) Strict consensus topology for the unweighted analysis; (B) strict consensus topology for the strongly downweighted analyses (k = 1 and k = 3), (C) strict consensus topology for the moderately downweighted analyses (k = 7 and k = 10); (D) strict consensus topology for the moderately to weakly downweighted analyses (k = 15, k = 20 and k = 50). Numbers indicate clades: (1) Geosaurinae, (2) Geosaurini, (3) E-clade. Complete strict consensus trees are provided in Material S3.

In all of the strict consensus trees, Torvoneustes jurensis is included in a polytomous clade with all terminal taxa assigned to Torvoneustes (Fig. 17). This clade forms a polytomy with the E-clade and Purranisaurus potens. In the unweighted analysis, this group made of Torvoneustes, the E-clade and P. potens forms a polytomy with Ty. lythrodectikos and a clade consisting of Geosaurina + Plesiosuchina (Fig. 17A). These taxa form together the Geosaurini. In the strongly downweighted analyses (k = 1 and k = 3), “M.” westermanni and “M.” casamiquelai assume more basal positions outside of geosaurines (Fig. 17B). Tyrannoneustes lythrodectikos is sister group to the E-clade, which corresponds to the ‘subclade T’ of Foffa, Young & Brusatte (2018). Within Geosaurina, I. melkshamensis and G. lapparenti switch positions. The moderately downweighted analyses (k = 7 and k = 10) result in a topology overall consistent with that of the strongly downweighted analyses, except that “M.” westermanni and “M.” casamiquelai regain a basal position among geosaurines (Fig. 17C). In the moderately to weakly downweighted analyses (k = 15, k = 20 and k = 50), I. melkshamensis and G. lapparenti switch back to the positions they have in the strict consensus of the unweighted analysis (Fig. 17D). Overall, the results of our parsimony analyses are consistent with those of Young et al. (2020b) and Abel, Sachs & Young (2020), except our strict consensus for the unweighted analysis that is less well resolved (subclade T only recovered by the weighted analyses).

To improve the resolution of relationships, unstable taxa were pruned a posteriori from the consensus trees to produce a maximum agreement subtree for each of the parsimony analysis (see Material and methods). For the unweighted analysis, a total of 13 OTUs, including eight geosaurines (“Metriorhynchus” brachyrhynchus, Tyrannoneustes lythrodectikos, Geosaurus lapparenti, Purranisaurus potens, Druegendorf merged, English rostrum, Torvoneustes sp., Torvoneustes mexicanus), are pruned from the original set of 180 OTUs. In contrast, 33 OTUs, including 12 geosaurines (“Metriorhynchus” brachyrhynchus, Neptunidraco ammoniticus, Purranisaurus potens, Druegendorf merged, English rostrum, Mr. Passmore’s specimen, Chouquet cf. hastifer, Torvoneustes sp., Torvoneustes mexicanus, Geosaurus grandis, Geosaurus giganteus, and Ieldraan melkshamensis or Geosaurus lapparenti), are pruned for the weighted analyses. The four new topologies obtained for relationships within Geosaurinae are presented in Fig. 18 and the complete pruned consensus trees are available in Material S4.

Figure 18 Phylogenetic placement of MJSN BSY008-465 (Torvoneustes jurensis sp. nov.) within Geosaurinae after pruning of unstable taxa.

(A) Maximum agreement subtree for the unweighted analysis; (B) maximum agreement subtree for the strongly downweighted analyses (k = 1 and k = 3); (C) maximum agreement subtree for the moderately downweighted analyses (k = 7 and k = 10); (D) maximum agreement subtree for the moderately to weakly downweighted analyses (k = 15, k = 20 and k = 50). Numbers indicate clades: (1) Geosaurinae, (2) Geosaurini, (3) E-clade. Complete pruned consensus trees are provided in Material S4.

All maximum agreement subtrees suggest that Torvoneustes sp. and Torvoneustes mexicanus are unstable taxa. This is probably the result of the partial nature of these terminals represented by single specimens consisting of an incomplete occipital region and a portion of rostrum, respectively. The pruning of these taxa reveals the internal relationships of the Torvoneustes clade. Torvoneustes coryphaeus is recovered as the most basal taxon in a sister group relationship with a clade consisting of cf. Torvoneustes and To. jurensis + To. carpenteri (Fig. 18). From a more general perspective, the internal relationships of the E-clade and Geosaurina are also identified as unstable.

The Bayesian analysis results in a resolved, but poorly supported tree (Fig. 19). Torvoneustes jurensis is resolved as the sister taxon of To. carpenteri. These two species form the most derived clade within the clade Torvoneustes. Torvoneustes mexicanus is found as the sister taxon of To. jurensis + To. carpenteri, whereas To. coryphaeus appears as the most basal unit. Most nodes in the Torvoneustes clade and the E-clade are weakly supported. The Bayesian topology for Geosaurini is similar to the one obtained by Young et al. (2020b), with only two exceptions: (1) the position of cf. Torvoneustes and Torvoneustes sp. are switched; (2) To. mexicanus and To. carpenteri are no longer sister taxa. In our analysis, the node supports within subclade T is slightly lower, which can be explained by the inclusion of To. jurensis as a new terminal. the complete maximum compatibility is are available in Material S5.

Figure 19 Phylogenetic placement of MJSN BSY008-465 (Torvoneustes jurensis sp. nov) within Metriorhynchidae in the Bayesian analysis.

Numbers in red represent node support values. Numbers on the tree branches indicate clades: (1) Metriorhynchidae, (2) Geosaurinae and (3) Geosaurini. Complete tree is provided in Material S5.

The different phylogenetic analyses performed as part of the present study all consistently find To. jurensis (MJSN BSY008-465) nested within a Torvoneustes clade, supporting our identification. Both the Bayesian analysis and the maximum agreement subtrees of the parsimony analyses support a close relationship between To. jurensis and To. carpenteri.

Discussion

MJSN BSY008-465 assigned to Geosaurinae

The absence of mandibular fenestrae, the orbits facing laterally and overhung by the prefrontals, and the absence of osteoderms (despite the preservation of numerous postcranial remains) unambiguously place MJSN BSY008-465 among Metriorhynchidae (Fraas, 1901, 1902; Andrews, 1913; Young et al., 2010). In this section, we discuss the assignment of this specimen to Geosaurinae.

The dental characteristics of the Late Jurassic metriorhynchids allow to discriminate the Geosaurini from the Metriorhynchinae. The latter usually have smooth to faintly ornamented teeth, uncarinated or with low, non-serrated carinae, whereas Geosaurini have smooth to heavily ornamented teeth with high, serrated carinae (Table 1). The presence of prominent serrated carinae appears to be restricted to the Geosaurini tribe (Andrade et al., 2010; Young et al., 2011). Metriorhynchids genera can even be identified based on teeth only and within Geosaurini, teeth can be used for species identification, (Young & Andrade, 2009; Andrade et al., 2010; Schaefer, 2012; Young et al., 2013a, 2013b; Barrientos-Lara et al., 2016; Foffa et al., 2017; Foffa, Young & Brusatte, 2018; Schaefer et al., 2018; Madzia et al., 2021). MJSN BSY008-465 shares with Torvoneustes the presence of conspicuous apicobasal ridges on the first two-thirds of the crown shifting to an anastomosed pattern on the apex, as well as the bending of the enamel ridges toward the carinae. The teeth of some Metriorhynchinae (Cricosaurus spp., Maledictosuchus nuyivijanan) resemble those of Torvoneustes with conspicuous apicobasal ridges on the first two-thirds of the crown, but in their case the apex is smooth and the carinae are low and non-serrated (Sachs et al., 2019; Table 1).

Derived geosaurines, such as Plesiosuchus, Dakosaurus, Torvoneustes, Geosaurus, Purranisaurus and the E-clade, show an extreme reduction in interalveolar space associated with a reduction of the tooth count and a moderate enlargement of the teeth (Young et al., 2012b, 2013a, 2013b; Herrera, Gasparini & Fernández, 2015; Abel, Sachs & Young, 2020). Metriorhynchines, including those with a low tooth count such as ‘Cricosaurus’ saltillensis (see below), have large and variable interalveolar spaces (Buchy, Young & Andrade, 2013; Young et al., 2020a; Herrera, Aiglstorfer & Bronzati, 2021; Herrera, Fernández & Vennari, 2021). The only exception is Gracilineustes leedsi in which reduced interalveolar spaces are associated with a high tooth count (+30 per maxilla; Young et al., 2013b). MJSN BSY008-465 has reduced interalveolar spaces associated with moderately enlarged teeth, which corresponds to the condition in derived geosaurines. In addition, the teeth of MJSN BSY008-465 are on average larger than the typical height observed for metriorhynchine teeth, which are usually shorter than two centimeters (Wilkinson, Young & Benton, 2008; Herrera, Fernández & Vennari, 2021).

The tooth count is often used to differentiate geosaurines from metriorhynchines (Young et al., 2013b), but it should be noted that the absolute tooth count is known only in a limited number of species (Table 1). Geosaurini are usually considered to have 16 or less teeth per maxilla (Cau & Fanti, 2011). With this in mind, the estimated tooth count for MJSN BSY008-465 (17–18 or more, see above) may seem high for a Geosaurini, especially when some non-rhacheosaurin metriorhynchines, such as “C.” saltillensis and “C.” macrospondylus, present comparable tooth counts (Table 1; Buchy, Young & Andrade, 2013; Aiglstorfer, Havlik & Herrera, 2020). However, this relatively low tooth count in some rhacheosaurins seems to be linked to a pronounced shortening of the skull. On the other hand, it appears that the tooth count is poorly estimated in Torvoneustes because no complete maxilla is known, and some species have a comparable tooth count as MJSN BSY008-465. For example, To. coryphaeus is estimated to have up to 19 alveoli per maxilla (Young et al., 2013b), which also falls into the range of other geosaurines such as Chouquet’s “Metriorhynchus” cf. hastifer and its at least 20 maxillary teeth (Lepage et al., 2008). Therefore, the tooth count for Torvoneustes is maybe underestimated for the moment based on the available material. It is also possible that the tendency toward the great reduction in the number of teeth is restricted to the clade uniting Geosaurina, Dakosaurina, and Plesiosuchina. In any case, it appears that tooth count, as a tool for identification, should be handled with care, especially when based on estimations.

MJSN BSY008-465 shares with Geosaurinae the presence of an acute angle of about 60° between the medial and lateral processes of the frontal. This angle is closer to 90° in most Metriorhynchinae, to the exception of Cricosaurus and Maledictosuchus in which this angle is around 45°–50° (Wilkinson, Young & Benton, 2008; Cau & Fanti, 2011; Buchy, Young & Andrade, 2013; Parrilla-Bel et al., 2013; Foffa & Young, 2014). The new specimen described herein shares several additional cranial and mandibular features with the macrophagous predators of the Geosaurini tribe: an inflection point of the prefrontals relative to the skull midline of 70° or less; a high glenoid fossa and retroarticular process; a strongly expressed surangular-dentary groove (Young & Andrade, 2009; Young et al., 2012b, 2013b; Foffa & Young, 2014). In addition to some dental characteristics discussed above, MJSN BSY008-465 also has some cranial features that may recall the metriorhynchine Cricosaurus. MJSN BSY008-465 notably has a smooth and unornamented cranial surface, but this is also the case of D. andiniensis and P. potens (Pol & Gasparini, 2009; Herrera, Gasparini & Fernández, 2015). The frontal of MJSN BSY008-465 differs in shape from that of other geosaurines and somewhat resembles that of “C.” saltillensis (Buchy, Young & Andrade, 2013), but there is a great diversity of frontal shapes among metriorhynchids (Foffa & Young, 2014, fig. 10; Herrera, 2015). Despite these few similarities, MJSN BSY008-465 lacks some cranial characters that are typical of Cricosaurus, such as the presence of a bony septum on the premaxillary and the presence of reception pits on the maxilla (as seen in C. bambergensis and C. albersdoerferi; Sachs et al., 2019, 2021). Therefore, the craniomandibular characters, like the dental characters, indicate that MJSN BSY008-465 should be assigned to Geosaurinae and suggest that the resemblances with Cricosaurus are only superficial.

The total body length of MJSN BSY008-465 is estimated to be around four meters (De Sousa Oliveira et al., 2023), which exceeds the sizes typically measured and estimated for Rhacheosaurus (157 cm), Cricosaurus (200 cm), and Geosaurus (270 cm), but falls in the range of large-bodied geosaurines such as Suchodus durobrivensis Lydekker, 1890 (410 cm), D. andiniensis (430 cm); To. coryphaeus (370 cm) and To. carpenteri (400–470 cm) (Young et al., 2010, 2019). Because this specimen is one of the few metriorhynchids found with a significant part of its postcranium (see also Sachs et al., 2019; Le Mort et al., 2022), some remarks relative to the assignment of the specimen should be made also on this part of the skeleton.

Geosaurinae (Neptunidraco ammoniticus, “M.” brachyrhynchus, Ty. lythrodectikos, To. carpenteri, Geo. lapparenti, D. maximus) and MJSN BSY008-465 share a centrum length subequal to centrum width on cervical vertebrae, whereas in other Metriorhynchidae (Thalattosuchus superciliosus, Rhacheosaurus gracilis, C. araucanensis, C. suevicus, C. bambergensis, C. albersdoerferi) the centrum is shorter than wide (see character 423 of the phylogenetic matrix; Parrilla-Bel & Canudo, 2015). In MJSN BSY008-465, the neural length spine of the dorsal vertebrae is about half the length of the centrum and its dorsal margin is rounded. This is markedly different from C. suevicus and C. albersdoerferi in which the neural spine of the dorsal vertebrae is wide and rectangular with a flat dorsal margin and subequal in length to the length of the centrum (Sachs et al., 2021). The centrum of the dorsal vertebrae is also distinctly longer than high in C. albersdoerferi and Cretaceous metriorhynchids (Sachs, Young & Hornung, 2020; Sachs et al., 2021), whereas the centrum length is subequal to its height in MJSN BSY008-465 as in E-clade member PSHME PH1 and N. ammoniticus (Cau & Fanti, 2011; Abel, Sachs & Young, 2020).

In metriorhynchids there is a drastic reduction of the length of the tibia and fibula compared to the femur (Fraas, 1902; Andrews, 1913; Foffa et al., 2019). The tibia of MJSN BSY008-465 is not preserved, but the fibula is usually subequal or slightly longer than the tibia in metriorhynchids and can therefore be used as a proxy (Sachs et al., 2019). As preserved, knowing that each bone is missing parts of the articular heads, the fibula of MJSN BSY008-465 is about 30% of the femoral length, which corresponds to the proportions usually observed in metriorhynchids. Members of the tribe Rhacheosaurini appear to be an exception because their tibia is less than 30% of the femur length (Andrews, 1913; Wilkinson, Young & Benton, 2008; Young & Andrade, 2009; Cau & Fanti, 2011; Foffa et al., 2019). However, this character (see #518 in the phylogenetic matrix; Supplementary data in Foffa et al., 2019) can only be scored in a small number of metriorhynchids, and its repartition should be further investigated, especially in derived geosaurines such as Dakosaurus. Although not as diagnostic as the dental and cranial characters, the postcranial characters of MJSN BSY008-465 tend to suggest an affinity of this specimen with geosaurines rather than with non-metriorhynchine metriorhynchid and to exclude a relationship with rhacheosaurins such as Cricosaurus.

Taxonomic diversity in the geosaurine genus Torvoneustes

The genus Torvoneustes is a member of Geosaurinae and is currently represented by three valid species: the type species To. carpenteri from the upper Kimmeridgian, a skull heavily crushed and not described in many details, and some postcranial remains from a second specimen (Grange & Benton, 1996; Wilkinson, Young & Benton, 2008; Andrade et al., 2010); To. coryphaeus from the lower Kimmeridgian, a 3D preserved skull missing the anterior part of the rostrum (Young et al., 2013b); and To. mexicanus, likely from the Kimmeridgian, represented by a piece of a rostrum (Barrientos-Lara et al., 2016). Other specimens were referred to the genus and include several isolated teeth form the Oxfordian (BRSMG Cd5591, Cd5592; CAMSM J.13305J, 13309, J.13310, YORYM:2016.306–2016.309; OUMNH J.52428, J.47587a, J.47560; Foffa, Young & Brusatte, 2018); three specimens referred to Torvoneustes sp., MJML K1707 from the Upper Kimmeridgian; OUMNH J.50061 and OUMNH J.50079-J.50085 from the lower Tithonian (Young et al., 2019); cf. Torvoneustes (MANCH L6459) from the middle Oxfordian (Young, 2014), and Torvoneustes? (NHMW 2020/0025/0001) an isolated tooth crown from the upper Valanginian. All specimens are from England, except To. mexicanus and Torvoneustes?, which are from Mexico and Czech Republic respectively (Table 1).

Based on the three valid species, the genus Torvoneustes is defined by the following characteristics (Wilkinson, Young & Benton, 2008; Andrade et al., 2010; Young et al., 2013b; Barrientos-Lara et al., 2016): great reduction of the interalveolar space; acute angle (around 60°) between the medial and lateral processes of the frontal; inflexion point on the lateral margin of the prefrontals directed posterolaterally at an angle of ~70° from the anteroposterior axis of the skull; circular to subcircular tooth cross section; carina formed by a keel and a contiguous row of poorly defined microscopic denticles difficult to observe even under SEM observation; conspicuous enamel ornamentation consisting of subparallel apicobasal ridges on the first two thirds of the crown shifting to short, low relief tubercles on the apex. The new specimen described herein closely follows this definition, but also presents significant differences with each of the recognized species.

The frontal of MJSN BSY008-465 is shaped differently than those of To. carpenteri and To. coryphaeus. In To. carpenteri, the frontal is shorter than in To. coryphaeus and MJSN BSY008-465. Torvoneustes coryphaeus has an ornamented frontal while in To. carpenteri and MJSN BSY008-465 the frontal is smooth. Finally, MJSN BSY008-465 is characterized by a clear angle between the anterior and posterolateral processes of the frontal, at the meeting point of the frontal, nasals and prefrontals. In the aforementioned two species, there is no visible angle and the processes are aligned in an almost straight line. Variation in shape of the frontals in the genus Torvoneustes is not well known, as only two described specimens preserve this element in addition to the new material described herein. Within metriorhynchids, we can note the great interspecific variation in the shape of the frontal (Foffa & Young, 2014). For example in Cricosaurus, there is a great variation of the frontal shape, as seen in C. araucanensis (Herrera, 2015).

MJSN BSY008-465 cannot be compared with Torvoneustes sp. (MJML K1707). The latter consists of an incomplete occipital region and this part is completely lost in our specimen. However, MJSN BSY008-465 can be distinguished from all other English specimens. Torvoneustes jurensis differs from the type species To. carpenteri based on the following characters: smooth maxillae without grooves; anterior process of the frontal reaching the anterior margin of the prefrontals; posterolateral edges of the prefrontals lacking “finger-like” projections; teeth slenderer, more curved, and with a sharp apex. MJSN BSY008-465 also differs from To. coryphaeus in having: a smooth skull; rounded posterolateral edges of the prefrontals (no acute angle); slender teeth with sharp apex; tooth ornamentation touching the carina in the upper part of the crown. Isolated Torvoneustes sp. teeth from the UK (BRSMG Cd5591, Cd5592; CAMSM J.13305J, 13309, J.13310, YORYM:2016.306–2016.309; OUMNH J.52428, J.47587a, J.47560, J.50061, J.50079-J.50085) preserve tooth crowns and roots very similar to To. carpenteri, As noted above, MJSN BSY008-465 has slender teeth. Finally, MJSN BSY008-465 is different from cf. Torvoneustes MANCH L6459 by its smooth cranium.

Torvoneustes mexicanus is only known by a single specimen that consists of a fragment of snout with preserved teeth. The species diagnosis is based only on the teeth with the following characters: conical, lingually curved, bicarinate, and more slender than in other Torvoneustes species; sharp apex; microziphodont condition with well-defined isomorphic denticles; crown enamel ornamentation consisting of apicobasally aligned ridges on the basal two-thirds of the crown and shifting to short drop-shaped tubercles meeting the carina on the apex (Barrientos-Lara et al., 2016). On macroscopic observation, the teeth of MJSN BSY008-465 and To. mexicanus are very similar, but they differ on microscopic features. The denticles of To. mexicanus are well defined, regular in shape, size and distribution with a denticle basal length of about 142 µm and a denticle density (#denticles/5 mm) of 30 (Barrientos-Lara et al., 2016). The crown height seems to range between 1 and 2.5 cm. In MJSN BSY008-465, the basal length and distribution of denticles are irregular; on the tooth total length, the base of the denticles can vary from 120 to 200 µm in length with an average of 160 µm measured on four different teeth; denticles are also more densely packed in the upper middle of the tooth with a density reaching 40 while they can drop to 30 on the basal most part of the carina and the tooth apex. These observations are homogenous on the four observed teeth. It is to be noted however that denticles are sometimes hard to discern, because of their shape and size but also due to areas where they are worn or where the carina is broken. We compared these results with measurements based on formerly published SEM photographs of teeth of To. carpenteri (Young et al., 2013a). In this species, the denticle basal length varies between 120 and 220 µm, with most measured denticles having a basal length between 160 and 200 µm. Unfortunately, denticle density cannot be determined. Again based on published SEM photographs (Madzia et al., 2021), the denticles of Torvoneustes? (NHMW 2020/0025/0002) range in size from 200 to 270 µm in basal length and have a density of 19.

Previous studies showed that the denticle density is variable between geosaurines species such as Dakosaurus and Geosaurus (Andrade et al., 2010). Measurements took in the middle of the carina gives the following densities for the microziphodont specimens: 28.1 in Geosaurus indet. (NHM R.486), and 33.3 for the mesial carina and 41.7 for the distal carina in G. grandis. These results suggest possible interspecific variations in denticles density for a same tooth morphotype among geosaurines and that these variations should not be overlooked for systematic purposes (Andrade et al., 2010). In this study however, the density of denticles is only measured on one tooth. The intraspecific and the individual variation, which were documented in other studies on crocodylomorphs teeth (Prasad & de Lapparent de Broin, 2002), were not explored in this case. However, the observations on MJSN BSY008-465, in addition to previous studies on metriorhynchid teeth, support the idea that microscopic dental characteristics in metriorhynchids have a potential to be used in systematics and should be further investigated.

From the above discussion and considering that To. mexicanus is only known by a very incomplete specimen of uncertain stratigraphical origin, it seems reasonable to conclude that MJSN BSY008-465 represents a different species (Table 3), which we name Torvoneustes jurensis. Future discoveries of more complete fossil specimens of To. mexicanus will allow a better understanding of the differences between these species.

Table 3 Comparison of cranial features within the genus Torvoneustes.

	Skull ornamentation	Frontal	Distinct angle at the
frontal-prefrontal-nasal contact	Tooth count
per maxilla (estimated)	Teeth morphology	False serration	Denticles size	Denticles density (#denticules/5 mm)	
Torvoneustes carpenteri
(Grange & Benton, 1996; Wilkinson, Young & Benton, 2008)	Smooth with
ornamentation on the maxillae	Does not reach the
anterior end of the prefrontal	No	14	Robust, blunt apex	Yes	Irregular in size.
BS: 120–220 µm	–	
Torvoneustes coryphaeus
(Young et al., 2013b)	Ornamented	Reach the
anterior end of the prefrontal	No	17–19	Robust, blunt apex	No	–	–	
Torvoneustes mexicanus
(Barrientos-Lara et al., 2016)	–	–	–	–	Slender, sharp apex	Yes	Regular in size.
BS: 142 µm	30	
Torvoneustes jurensis sp. nov.	Smooth	Reach the
anterior end of the prefrontal	Yes	Up to 18	Slender, sharp apex	Yes	Irregular in size.
BS: 120–200 µm	30–40	
Note:

Comparative table of currently named species of Torvoneustes and their diagnostic cranial features identified in To. jurensis. False serration definition and denticle density methodology follow Andrade et al. (2010) (see description for more details). BS, Base length.

Macroevolution trends in Torvoneustes

Young et al. (2013b, 2019) discussed macroevolutionary trends in the genus Torvoneustes, we here include the new taxon To. jurensis within these putative trends and discuss how its addition strengthens or contradicts them. The first potential trend noted by Young et al. (2013b, 2019) is a reduction of the maxillary tooth count with time from “relatively high” in cf. Torvoneustes (MANCH L6459; Young et al., 2019) and 17–19 in To. coryphaeus (Young et al., 2013b), to 14 in To. carpenteri (Wilkinson, Young & Benton, 2008). However, it should be noted that no complete maxilla is known for any specimen referred to Torvoneustes, so these tooth counts represent estimations (see above). Another trend noted in Torvoneustes is a decrease in dermocranial external ornamentation, with the upper Kimmeridgian To. carpenteri having a smoother skull than the older representatives To. coryphaeus and cf. Torvoneustes MANCH L6459 (Grange & Benton, 1996; Wilkinson, Young & Benton, 2008; Young et al., 2013b; Young, 2014).

Concerning the tooth morphology, the following macroevolutionary trends were proposed: increasing enamel ornamentation; blunter crown apices; tooth crown losing the lingual curvature; and crown cross section becoming subconical (Young et al., 2019). These trends are interpreted to be linked to an increasingly durophagous diet (Young et al., 2013b, 2019; Foffa et al., 2018). Torvoneustes coryphaeus, To. carpenteri, Torvoneustes? (NHMW 2020/0025/0001) and Torvoneustes sp. (OUMNH J.50061 and OUMNH J.50079-J.50085) fit relatively well into this proposed evolutionary trend. However, that is not the case of To. mexicanus which has more slender teeth than To. coryphaeus and To. carpenteri, as well as more curved teeth than the latter. The teeth of To. mexicanus are also sharper than those of To. coryphaeus, To. carpenteri and Torvoneustes sp. OUMNH J.50061 and OUMNH J.50079-J.50085.

The acquisition of false serration (false ziphodont dentition; Prasad & de Lapparent de Broin, 2002; Young & Andrade, 2009; Andrade et al., 2010) is another macroevolutionary trend proposed for Torvoneustes (Young et al., 2019). Torvoneustes coryphaeus is the only species lacking the false ziphodont dentition and is also the oldest specimen whose teeth are known (Table 3). The other specimens preserving teeth (To. carpenteri; Torvoneustes sp., OUMNH J.50061 and OUMNH J.50079-J.50085; Torvoneustes?, NHMW 2020/0025/0001; To. mexicanus) all present the false ziphodont condition. Therefore, the authors propose the acquisition of the false ziphodont condition in all species younger than To. coryphaeus (Young et al., 2019).

The discovery of the occipital region of a Torvoneustes specimen of great size from the Tithonian of England led (Young et al., 2019) to propose an increase of body size as a possible evolutionary trend within Torvoneustes. This specimen is estimated to be around 6 m long while other Torvoneustes specimens are estimated to be between 3.70 and 4.70 m long (Young et al., 2011, 2019). And finally, Young et al. (2019) also suggested that the increase in length of the suborbital fenestrae leading to an enlarged pterygoid musculature and the ventralization of basioccipital tuberosities would be another evolutionary trend of Torvoneustes. However, because this part of the cranium is not preserved in To. jurensis, this trend will not be further discussed below.

Torvoneustes jurensis fits relatively well with some of the aforementioned macroevolutionary trends. First, To. jurensis is younger than To. coryphaeus and indeed presents the false ziphodont condition on its teeth. However, it should be noted that in the current state of knowledge the distribution of this character does not per se corresponds to an evolutionary trend, especially considering the observation of this character among Oxfordian isolated teeth (Foffa, Young & Brusatte, 2018), older than To. coryphaeus. Therefore, false ziphodont condition may be a synapomorphy uniting Torvoneustes species but lost in To. coryphaeus.

Torvoneustes jurensis, which is from the early late Kimmeridgian, has an even smoother cranium than To. carpenteri. Dermocranial ornamentation appears early during ontogeny in crocodylomorphs, usually developing in specimens with a skull longer than 200 mm (de Buffrenil, 1982; de Buffrénil et al., 2015). This indicates that the smooth cranium of To. jurensis is not linked to its ontogenic stage and fits into the evolutionary trend proposed by Young et al. (2019). Within Thalattosuchia, the trend toward smoother dermocranial bones is believed to improve hydrodynamic efficiency (Young et al., 2013b) and overall follows the idea that pelagic species have less ornamented skulls than semi-aquatic one (comparing metriorhynchids and pelagic teleosauroids to non-pelagic teleosauroids for example; Clarac et al., 2017; Foffa et al., 2019). This trend is also present in Dakosaurus with D. andiniensis, the geologically younger species, showing smoother cranial bones than D. maximus (Pol & Gasparini, 2009; Young et al., 2012a). However, the functional role of bone ornamentation remains controversial in crocodylomorphs (de Buffrénil et al., 2015). Clarac et al. (2017) presented evidence that the evolution of ornamentation in pseudosuchians is influenced by both natural selection and Brownian motion. The study shows that heavy ornamentation is present in pseudosuchians with semi-pelagic lifestyle and linked to basking for animals with low mobility, these results are backed by findings from Pochat-Cottilloux et al. (2022). Therefore, the loss of ornamentation in Thalattosuchia may rather be linked to an increasingly pelagic lifestyle than directly to hydrodynamic efficiency. The reduction of ornamentation of the cranium and osteoderms was also observed by Foffa et al. (2019) in teleosauroids and similarly linked it to adaptation to pelagic lifestyle.

The description of To. jurensis contradicts some aspects of the other proposed evolutionary trends. Torvoneustes jurensis presents slenderer and sharper teeth than the other Torvoneustes species, except To. mexicanus. In their description, Barrientos-Lara et al. (2016) raised the question of whether the slender and sharp teeth of To. mexicanus could be linked to ontogeny, but the lack of data to characterize ontogenic changes within Torvoneustes teeth led them to consider the differences between To. mexicanus and other Torvoneustes as a specific characteristic. The teeth of To. jurensis are very similar to those of To. mexicanus. However, the skull length of the holotype of To. jurensis is similar to that of the holotype of To. carpenteri. Their body length is estimated to be close to 4.0 m for To. jurensis and between 4.0 and 4.70 m for To. carpenteri (Grange & Benton, 1996; Wilkinson, Young & Benton, 2008; Young et al., 2011; De Sousa Oliveira et al., 2023). Therefore, ontogeny cannot explain the differences in tooth morphology between these species. The sharp and slender teeth of To. jurensis and To. mexicanus may instead represent a diverging tooth morphotype within the genus. It might indicate that To. mexicanus and To. jurensis are less specialized than species with more robust teeth. They might be opportunist feeders with durophagous tendencies. This would be consistent with the Kimmeridgian environment of the Jura platform: Torvoneustes jurensis was found in a carbonate platform environment where remains of teleosauroids (Sericodon jugleri von Meyer, 1845; Proexochokefalos cf. bouchardi and another durophagous genus Machimosaurus hugii Meyer, 1837) are abundant, as well as many coastal marine turtles (Thalassochelydia Anquetin, Püntener & Joyce, 2017) and fishes with hard scale (e.g., Scheenstia sp., Fig. 20).

Figure 20 Life reconstruction of Torvoneustes jurensis MJSN BSY008-465 in the Kimmeridgian environnement of Jura.

Artwork by SDSO.

The maxillary tooth count of To. jurensis is estimated to be at least 16 or 17, but is probably higher. Therefore, it seems like the reduction of maxillary tooth count is not a homogenous trend in the genus. However, it should be stressed once more that no definitive maxillary tooth count is known at the moment for any specimen referred to Torvoneustes. It is then possibly too early to conclude on any trend for this character.

Regarding the increase in body size, it should be noted that crocodilians continue to grow well into adulthood (Sebens, 1987; Grigg & Kirshner, 2015) and that only a handful of specimens are known for Torvoneustes. In these circumstances, any conclusion on size evolutionary trends must therefore be taken with care. As noted above, the holotype specimens of To. carpenteri and To. jurensis are roughly of comparable total length, which could agree with the proposed evolutionary trend as the two species are roughly of the same age. However, many isolated teeth showing similar characteristics as those of MJSN BSY008-465 were found in the same stratigraphical layers during the excavation on the A16 highway, along with teeth of Sericodon, Proexochokefalos, Machimosaurus and Dakosaurus (Schaefer, 2012; Schaefer et al., 2018): teeth with sub-circular to ovoid cross-section, bicarinate with micro-ziphodont condition, enamel ornamentation composed of sub parallel apicobasal ridges on the basal two thirds of the crown shifting into low relief drop-shaped ridges forming an “anastomosed pattern” on the remaining upper third of the crown. These teeth show a great variation of height and base length (Schaefer, 2012; L. Girard, 2021, personal observation). One of these teeth in particular (MJSN TCH007-91) shows a base length 31% longer than the largest tooth associated with MJSN BSY008-465. The great variation in crown size of isolated teeth indicates that specimens of various sizes (and potentially ages) visited the area, including specimens significantly larger than MJSN BSY008-465. Considering there is only a few Torvoneustes specimens known between the middle Oxfordian and the Tithonian, it seems premature to consider size increase as an evolutionary trend in Torvoneustes for the moment.

Conclusions

The holotype of Torvoneustes jurensis is the most complete skeleton of the genus and the first specimen to preserve both extensive cranial and postcranial material. This new species is distinct from the other species on the basis of cranial morphology, dental characters, and geographic distribution. The phylogenetic analysis tends to confirm these observations. The distinction between To. jurensis and To. mexicanus remains difficult due to the fragmentary nature of the Mexican specimen. While future discovery of more specimens of To. mexicanus might help to get a better understanding of the differences between these two taxa, the dental characters allow us to discriminate them as two distinct species. It is interesting to note also that the genera Cricosaurus and Dakosaurus are as well found in the Kimmeridgian deposits of Europe, Mexico and South America, but that they are represented by different species in each of these geographically distant areas (Buchy et al., 2006b; Buchy, Young & Andrade, 2013; Pol & Gasparini, 2009; Young et al., 2012b; Herrera, 2015; Herrera, Aiglstorfer & Bronzati, 2021). In addition, it seems that the Late Jurassic marine reptiles of the Mexican Gulf are represented by species that differ from the coeval European and South American fauna, with the notable exception of one specimen of Ophtalmosaurus (Buchy et al., 2006a; Buchy, 2007).

Supplemental Information

Supplemental Information 1 Character matrix derived from Young et al., 2020b, including 180 taxa coded for 574 characters.

Click here for additional data file.

Supplemental Information 2 Links to 3D models of the cranial bones of MJSN BSY008-465.

Each link provides access to 3D meshes used to reconstruct the skull and mandible of MJSN BSY008-465. Links to the reconstructed skull and mandible is also provided.

Click here for additional data file.

Supplemental Information 3 Complete strict consensus trees in the parsimony analysis.

Click here for additional data file.

Supplemental Information 4 Complete pruned consensus trees in the parsimony analysis.

Click here for additional data file.

Supplemental Information 5 Complete maximum compatibility tree in Bayesian analysis.

Click here for additional data file.

The authors would like to thank the Paleontology A16 team for the discovery and initial surface preparation of the specimen, Renaud Roch (Jurassica Museum) for his skillful complete extraction of the bones from the matrix, and Davit Vasilyan (Jurassica Museum) for the acquisition of the images from the digital microscope. We would also like to thank Corentin Jouault (Muséum National d’Histoire Naturelle) for his help with setting up Bayesian analysis and Mark Young (University of Edinburgh) for giving access to the character matrix and TNT script as well as for editing the manuscript. We thank the reviewers Pascal Abel (University of Tuebingen), Sven Sachs (Naturkunde-Museum Bielefeld) and Davide Foffa (Virginia Tech) for their comments that helped improve the quality of the manuscript, and Kenneth De Baets (University of Warsaw) for editing the second draft of the manuscript. We also thank the Willi Hennig Society for making TNT freely available.

Institutional abbreviations

BSPG Bayerische Staatssammlung für Paläontologie und Geologie, Munich, Germany

CAMSM Sedgwick Museum, Cambridge, UK

IGM Colección Nacional de Paleontología, Instituto de Geología, México, Mexico

MANCH Manchester Museum, Manchester, UK

NHMUK Natural History Museum, London, UK

NHMW Naturhistorisches Museum Wien, Vienna, Austria

MHNG Muséum d’Histoire Naturelle, Genève, Switzerland

MJML Museum of Jurassic Marine Life, Dorset, UK

MJSN Jurassica Museum (formerly Musée Jurassien des Sciences Naturelles), Porrentruy, Switzerland

NKMB Naturkunde Museum Bamberg, Bamberg, Germany

MUDE Museo del Desierto, Saltillo, Mexico

OUMNH Oxford University Museum of Natural History, Oxford, UK

SMNS Staatliches Museum für Naturkunde Stuttgart, Baden-Württemberg, Germany

YORYM Yorkshire Museum, York, UK

Additional Information and Declarations

Competing Interests

Author Contributions

Data Availability

New Species Registration

Jérémy Anquetin is an Academic Editor for PeerJ.

Léa C. Girard performed the experiments, analyzed the data, prepared figures and/or tables, authored or reviewed drafts of the article, and approved the final draft.

Sophie De Sousa Oliveira performed the experiments, prepared figures and/or tables, and approved the final draft.

Irena Raselli analyzed the data, authored or reviewed drafts of the article, and approved the final draft.

Jeremy E. Martin conceived and designed the experiments, authored or reviewed drafts of the article, and approved the final draft.

Jérémy Anquetin conceived and designed the experiments, authored or reviewed drafts of the article, and approved the final draft.

The following information was supplied regarding data availability:

The complete trees for the phylogenetic analysis, links to 3D models used for the skull reconstruction and the character matrix are in the Supplemental Files.

The 3D surfaces are available at Morphomuseum:

Sophie De Sousa Oliveira, Léa Girard, Irena Raselli and Jérémy Anquetin, 2023. M3#1037. DOI 10.18563/m3.sf.1037

Sophie De Sousa Oliveira, Léa Girard, Irena Raselli and Jérémy Anquetin, 2023. M3#1038. DOI 10.18563/m3.sf.1038

Sophie De Sousa Oliveira, Léa Girard, Irena Raselli and Jérémy Anquetin, 2023. M3#1039. DOI 10.18563/m3.sf.1039

Sophie De Sousa Oliveira, Léa Girard, Irena Raselli and Jérémy Anquetin, 2023. M3#1040. DOI 10.18563/m3.sf.1040

Sophie De Sousa Oliveira, Léa Girard, Irena Raselli and Jérémy Anquetin, 2023. M3#1041. DOI 10.18563/m3.sf.1041

Sophie De Sousa Oliveira, Léa Girard, Irena Raselli and Jérémy Anquetin, 2023. M3#1042. DOI 10.18563/m3.sf.1042

Sophie De Sousa Oliveira, Léa Girard, Irena Raselli and Jérémy Anquetin, 2023. M3#1043. DOI 10.18563/m3.sf.1043

Sophie De Sousa Oliveira, Léa Girard, Irena Raselli and Jérémy Anquetin, 2023. M3#1044. DOI 10.18563/m3.sf.1044

Sophie De Sousa Oliveira, Léa Girard, Irena Raselli and Jérémy Anquetin, 2023. M3#1045. DOI 10.18563/m3.sf.1045

Sophie De Sousa Oliveira, Léa Girard, Irena Raselli and Jérémy Anquetin, 2023. M3#1046. DOI 10.18563/m3.sf.1046

Sophie De Sousa Oliveira, Léa Girard, Irena Raselli and Jérémy Anquetin, 2023. M3#1047. DOI 10.18563/m3.sf.1047

Sophie De Sousa Oliveira, Léa Girard, Irena Raselli and Jérémy Anquetin, 2023. M3#1048. DOI 10.18563/m3.sf.1048

Sophie De Sousa Oliveira, Léa Girard, Irena Raselli and Jérémy Anquetin, 2023. M3#1049. DOI 10.18563/m3.sf.1049

Sophie De Sousa Oliveira, Léa Girard, Irena Raselli and Jérémy Anquetin, 2023. M3#1050. DOI 10.18563/m3.sf.1050

Sophie De Sousa Oliveira, Léa Girard, Irena Raselli and Jérémy Anquetin, 2023. M3#1051. DOI 10.18563/m3.sf.1051

Sophie De Sousa Oliveira, Léa Girard, Irena Raselli and Jérémy Anquetin, 2023. M3#1052. DOI 10.18563/m3.sf.1052

Sophie De Sousa Oliveira, Léa Girard, Irena Raselli and Jérémy Anquetin, 2023. M3#1053. DOI 10.18563/m3.sf.1053

Sophie De Sousa Oliveira, Léa Girard, Irena Raselli and Jérémy Anquetin, 2023. M3#1054. DOI 10.18563/m3.sf.1054

Sophie De Sousa Oliveira, Léa Girard, Irena Raselli and Jérémy Anquetin, 2023. M3#1055. DOI 10.18563/m3.sf.1055

The following information was supplied regarding the registration of a newly described species:

Publication LSID: urn:lsid:zoobank.org:pub:E8773387-2E64-4DF1-80ED-1A3A0C99D8BD

Torvoneustes jurensis: urn:lsid:zoobank.org:act:5DEFCF6F-D7EF-4711-9CB6-A7219F612ECB.

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
