# Peer review of "Description and phylogenetic relationships of a new species of Torvoneustes (Crocodylomorpha, Thalattosuchia) from the Kimmeridgian of Switzerland"

_PeerJ, doi:10.7717/peerj.15512_

## Round 0.1 · original submission · Major Revisions

Dear authors,

As the reviewers did not come to a common decision, I have decided to make a 'major revisions' recommendation. Note, the majority of comments come from reviewer three.

The reviewers comments can be broken down into four areas:
1. engagement with the literature.
2. comparisons with fossil specimens (other Torvoneustes species, particuarly the type species, and more comparisons with other metriorhynchids).
3. additional figures, and clearer versions of existing ones.
4. overstating of conclusions.

Reviewer three in particular goes through these issues in their comments and annotated PDF. Reviewer one mentions other metriorhynchid taxa to compare your specimen with (Purranisaurus potens and the 'E'-clade).

I look forward to receiving your revised manuscript.

·

Basic reporting

The manuscript text is overall comprehensible. The authors should make sure that taxon names are coherently and correctly used (e.g., Rhacheosaurini instead of “Rhacheosaurinae” or geosaurines instead of “geosaurids”).

In most cases, the authors show to be aware of the recent literature and cite it adequately. Sometimes the authors use taxa and specimen names that have since been disputed or updated in other works (i.e., “Steneosaurus”, “teleosaurids”, or “Solnhofen Cricosaurus”). I think this is mostly due to reliance on the character matrix provided by Young et al. (2021) and I suggest the authors to use a more recent version instead (Sachs et al. 2021; see also below). A Backgrounds section has not been submitted or at least not provided for the review.

The manuscript follows the PeerJ standards. All figures are necessary and of high quality. I just added one smaller note to Figs. 3 & 19. The necessary raw data has been provided.

Experimental design

The authors performed primary research based on novel material. The aim of the study is clear and the relevance for our understanding of metriorhynchid phylogenetics is highlighted. The methodology is sufficiently described and replicable.

Overall, the study follows the technical and ethical standards of the discipline. Regarding the phylogenetic analyses, the authors are encouraged to use the most recent version of their character matrix (to my knowledge, this is Sachs et al. 2021). The comparative anatomy part is rigorous; however, I strongly advise the authors to also compare their material to Purranisaurus potens (see Herrera et al. 2015). While not part of the Torvoneustes complex, P. potens has been in recent analyses always found to be a closely related species (contrary to Tyrannoneustes). Therefore, its relations to the new material should be further assessed. Potentially, also more comparisons with members of the “E-clade” might be reasonable (Lepage et al. 2008; Abel et al. 2020; Young et al. 2020b).

Validity of the findings

The findings are generally valid. Using an updated character matrix will likely lead to slightly different topologies but based on the material shown, I would not expect the new specimen to nest outside of Torvoneustes.

The conclusions are well-stated and link to the research question; however, I would suggest putting the part regarding metriorhynchid biogeography into the discussion instead.

Please see also the commented PDF for further details.

·

Basic reporting

'no comment'

Experimental design

'no comment'

Validity of the findings

'no comment'

Additional comments

Dear authors,
it was my pleasure to review your manuscript. The text is generally well written and concise. The figures are informative and the analysis includes all recently published taxa. I added some comments to the pdf file which may help to improve your manuscript and I'm looking forward to seeing it publication.
Best wishes,
Sven Sachs

·

Basic reporting

The scientific content of this work is clear, the borad take-away messages, structure of the manuscript are well thought and so are the figures (but additional views of the main bones and some comparative plates with other taxa (particulalry other Torvoneustes species would be beneficial). All raw data are provided (and I did appreciate the attached 3D data as well).

That said, the language (grammar and clarity) should be improved as it is ambiguous in places (see comments below, and annotated PDF). The engagement with the literature should be improved before this manuscript is published. Some outdated ideas, taxonomy and misconceptions are reported in the most general sections of the manuscript (Abstract, Introduction). It is important that these "gaps" should be addressed before the manuscript is published. Please see comments and suggestions to address these issues in the sections below and in the annotated PDF.

Experimental design

The research has a clear and well-defined objective that is relevant, and well pursued.
In places knowledge-gaps are outdated or overstated (see below and PDF) but this does not affect in any way the validity of the findings (in fact the findings do not address the "perceived knowledge gaps").
In my view the importance of this manuscript is the description of new material and new taxon of Torvoneustes, and the revision on how this fits in our understanding of the evolution of this genus. This is done meaningfully.

More comments below and in the annotated PDF.

Validity of the findings

I overall agree with the authors judgement that this specimen probably belongs to Torvoneustes and that it probably is a new species. All data have been provided, the results are robust.
In places the conclusions are overstated (flagged in the PDF), or need some clarifications in terms of additional comparisons/clearer figures.

I particularly would like to see more comparisons with other metriorhynchids in the description/discussion. Postcranial materials are available but have not often been described in detail. This is the perfect chance to do so, and it would be important to compare these new data with what is already available.

On a similar note, there are some features that should be figured: all those features that you use to distinguish To. jurensis from other species, should be figured in a comparative table with that of other species. You have done an excellent job showing the close up of the teeth, for example.
However, I think it would be beneficial if frontals, cranial ornamentation, and tooth morphology of different species of Torvoneustes were figured together in a comparative plate. It would make the comparisons immediately clear.

More comments below and in the annotated PDF.

Additional comments

Dear Editor and Authors,

This manuscript describes a new metriorhynchid specimen from the Late Jurassic deposits of Switzerland, and the authors put forward the case for a new species of Torvoneustes based on craniodental features. Overall, I think that the argument is valid, as I agree that the specimen in question has many key features of the genus (particularly in the teeth - e.g. the false serration, which are absent in To. coryphaeus), but also some key differences with other members of the genus (i.e. the prefrontal/frontal suture is very unusual). I concur that the combination of features warrant a new species within the genus. This argument is also supported by the phylogenetic analyses that the authors compiled.

There are some moderate revisions that I think should be addressed before the manuscript could be put forward for publication. They primarily concern the way the new information is reported, the extent of comparisons, and engagement with the literature.

1. Your most important issue concerns comparisons. More comparisons are needed.
The description of the new skeleton is appropriate, but would benefit from extensive comparisons with other known skeletons. As you mentioned skeletons this complete are not common, so it is important that they are appropriately documented and compared to others that are known. Particularly, I am puzzled by the scarce direct comparisons with overlapping elements of To. carpenteri, which is known from a fairly complete skeleton. Comparisons should probably be extended to the other members of Metriorhynchidae.
I would encourage the authors to add comparative plates as well if possible (see below).

2. The next most important item is Literature.
I have two remarks regarding the way the authors engaged with the literature.
Firstly , the engagement with recent literature should be expanded. This may sound as nit-picking but it actually spread some hard-to-kill misconceptions and outdated ideas in the manuscript.
For instance, in the abstract:
- metriorhynchids are described as being found in “mostly as isolated and fragmentary remains”. This seems like an overstatement to me: the fossil record of Thalattosuchia in general is not perfect, but there are many fairly complete skeletons in numerous collections);
- “the phylogenetic relationships within Metriorhynchidae are still heavily discussed”. The broad relationships within Metriorhynchidae have been well-established for 5-10-15 years now. The position of the main clades and sub-clades are fairly stable and so are their relationships.
- the Torvoneustes clade is called “poorly known”. In fact, Torvoneustes is probably one of the best known clades in Metriorhynchidae with 3-5 species and fairly complete skeletons within it!.
In the introduction, the authors introduce Thalattosuchia with a classification system that has been outdated for many years (i.e. Thalattosuchia = Teleosauridae + Metriorhynchidae being presented as the main clade in Thalattosuchia). The definition of Teleosauridae has changed (now replaced by Teleosauroidea), and so has our understanding of their ecological diversity (erroneously called “morphologically conservative”) (for both these issues see the recent works by Johnson et al. 2020, 2022). Similarly, there is a difference between Metriorhynchidae and Metriorhynchoidea (see Osi et al. 2018 https://peerj.com/articles/4668/ for instance) and there is some outdated definitions of Geosaurines vs Geosaurini that also require some fixing (I marked these issues in the PDF and suggested ways to address them).

Secondly, there are not many specimens referred to as Torvoneustes, so I’d expect that the authors would engage with all of these occurrences. At the moment the manuscript does not include a bunch of Torvoneustes sp. teeth (with false serrations) from the Oxfordian corallian Gap of the UK (Foffa et al. 2017 - https://www.app.pan.pl/archive/published/app63/app004552018.pdf). The inclusion of these specimens would not substantially change the content of the manuscript, but this is a matter of completeness. These specimens should be compared and are important in a broader context as they demonstrate the presence of a Torvoneustes taxon different from To. coryphaeus in the Oxfordian in the UK. They should be added to the discussion of the patterns within Torvoneustes.
I think that it would be beneficial for this work if the authors revised these sections after revising these issues with up-to-date literature. Due to the quantity of work done in Thalattosuchia in the last 20 years, these changes can be hard to follow, but it is important to do so to advance the field and find new research venues.


3. Please spend some time revising the grammar (spelling and sentence structure) and the clarity of the manuscript. Some sentences are hard to understand or ambiguous. I highlighted these issues in the manuscript, but I am myself not a native speaker so I may have missed some of them.

4. Figures. I find the figures overall well done and clear. I would still like to suggest some improvements. The most important ones are: every diagnostic feature should be clearly illustrated. So I would strongly recommend that the authors add a comparative plate with defining features of To. jurensis alongside the variation of that character in other Torvoneustes species. These would include the frontal, teeth, and close up of the ornamentation (which is not very visible at all in any of the figures) at least.
I would also encourage the authors to revise the figures by illustrating the standard anatomical elements in order (e.g. pmx, before nasal, before prefrontal). I left more comments in the annotated PDF as well.

5. The least important point concerns formatting.
The format of different sections is inconsistent. Please take some time to revise it and format consistently throughout the text.

Additional comments and minor (spelling, grammar, punctuation) issues can be found in the attached annotated PDF.

Overall, I wish to congratulate the authors and the whole team for finding, excavating, preparing and finally describing this important specimen, and for the very interesting manuscript. I will be looking forward to seeing this work published in due time. In the meantime I will be happy to answer any questions you may have over my review (email: davidefoffa@gmail.com or davidefoffa@vt.edu) if deemed appropriate by the editor and PeerJ editorial rules.

Many thanks,

Davide Foffa

---

## Round 0.2 · Minor Revisions

Dear authors,

The previous Academic Editor is not available so I have taken over handling your submission.

Your manuscript was sent back to the previous reviewer who suggested major revisions in round one. Thank you for addressing the comments of the reviewers which makes the manuscript easier to follow and of broader relevance. All relevant raw data and 3D data is provided as it should be. I would like to see this published, but there are some additional (mostly minor) points which need to be addressed:

1) Macroevolutionary trends: you discuss quite some putative/qualitative/potential (macro)evolutionary trends within this lineage. This add broader relevance and a nice additional dimension to the manuscript and should be kept but I feel you could express yourself a bit more carefully by adding putative/qualitative/potential to those potential trends. In the strict sense, an (macro)evolutionary trend should only be named as such when it has been quantitative and phylogenetically assessed in broader sample (as you seemingly agree when reading between the lines which should just be made slightly more explicit).

2) Comparative plate/figure/table: I agree with the reviewer that a plate/figure with extra comparative information concerning (other) Torvoneustes would be beneficial for the readers and to strengthen your arguments. I feel adding at least an extra table providing comparisons between all species in a more concise and systematic way would be highly beneficial to follow the discussion and more easily reproduce your assessments.

3) Typographical/formatting issues: there remains quite some typographical or formatting issues (e.g., Table 1). You can find these highlighted in the annotated pdfs of reviewer 3 and myself. Please take this opportunity to check the manuscript for such issues one more time before submission.

Please address these and all other points raised in reviews and annotated pdfs.

I look forward to receiving your revised manuscript.

·

Basic reporting

The main content of this manuscript was clear on the first submission and it still is. The structure and organisation of the main sections and figures are also suitably constructed. I thank the authors for taking on board some of my comments regarding the figures (order of the canonical views).
I still maintain that additional comparative plates (particularly with other Torvoneustes species) would be particularly beneficial for the readers and to strengthen their arguments. I am quite surprised that the authors decided against these relatively simple changes. As I mentioned these changes are not vital, and I recognize that this is ultimately their choice: it seems odd to leave out these very important comparisons.
All raw data are provided (and I appreciated the attached 3D data)

The language (grammar and clarity) have been improved from the first version, and so was the engagement with relevant literature. Because of these changes, outdated ideas and misunderstandings have been broadly fixed (although some others have been added - i.e. “exclusively pelagic Metriorhynchoidea” (in the Introduction)).
I thank the authors for going through the effort of addressing these issues.
There are still several grammatical mistakes (i.e., many typos, capital letters, spacing, imprecise or incorrect choice of word), and formatting inconsistencies: I tried to highlight as many as I could in the PDF. Please read through the text again and not only the parts I suggest modifying.

Experimental design

As I mentioned before, the importance of this manuscript is the description of new material and the new taxon of Torvoneustes, and I consider those parts valid. This is done meaningfully, and the revision has clarified some of the doubts I had.
Thankfully the comparisons were a little bit expanded, although more could have been done, especially with some elements (i.e. postcrania: of hundreds of available thalattosuchian femurs, ischia the authors decided to compare theirs with only a couple of metriorhynchid taxa).

Validity of the findings

I agree with the authors that this specimen probably belongs to Torvoneustes and likely is a new species.

Additional comments

Dear Editor and Authors,
This is the second time I revise this manuscript.
I was glad to see that the authors have engaged with the reviews (mine and of other reviewers), and I am pleased to say that this version of the manuscript has considerably improved.
I thank the authors for taking (most of) our comments on board and constructively working to better the manuscript, and I am particularly thankful for the renewed engagement with the literature.

Most of my comments are now minor and you can find them in the previous review sections or in the annotated PDF.

The main remaining issue, in my opinion, is grammatical accuracy. There are still many typos, odd choice of wording, and format inconsistencies.

Overall, I congratulate again the author and the whole team. I will be looking forward to seeing this work published in due time.
I will be happy to answer any questions you may have over my review (email: davidefoffa@gmail.com or davidefoffa@vt.edu) if deemed appropriate by the editorial team.

Many thanks,

Davide Foffa

---

## Round 0.3 · accepted · Accept

Thank you for addressing our suggestions including adding an additional table which makes the manuscript even clearer and easier to follow. I feel your manuscript can be accepted for publication pending some very minor changes suggested by the reviewer which can be integrated during the proofing phase. Please make sure the used terminology and assignments are consistent throughout the text and explicitly mention why you did not use the matrix of Young et al. (2020a) in the main text (e.g., I would suggest using or modifying the sentence you used in the previous rebuttal letter). Please make sure to address all points raised by the reviewer. I look forward to seeing your manuscript published.

·

Basic reporting

Dear Dr. De Baets,

I thank you for asking me to review this updated manuscript. I am pleased that the authors included most of my suggested changes. I think the paper has been greatly improved and I suggest acceptance after some final edits. Those are as follows:

Line 59, regarding Middle Jurassic
My original comment referred only to Geosaurini. To my knowledge, no rhacheosaurins are known from the Middle Jurassic.

Line 124-125
I accept your decision to use the matrix of Young et al. 2020a. However, I would suggest to also include here a sentence on why you did not use a newer matrix.

Line 572
Change "teleosaurid" in "machimosaurid" or "teleosauroid" (you also call Machimosaurus a machimosaurid later in the text).

Lines 964-965
"C." saltillensis is within Rhacheosaurini, "C." macrospondylus likely not (see my previous review).

Table 1
"M." palpebrosus is a rhacheosaurin. Species name in M. geoffroyi/brevirostris in lowercase. Maledictosuchus is outside of Rhacheosaurini.

Yours sincerely,
Pascal Abel

Experimental design

No comment.

Validity of the findings

No comment.

Additional comments

No comment